# The Tryptophan-Kynurenine Metabolic System Is Suppressed in Cuprizone-Induced Model of Demyelination Simulating Progressive Multiple Sclerosis

**DOI:** 10.3390/biomedicines11030945

**Published:** 2023-03-20

**Authors:** Helga Polyák, Zsolt Galla, Nikolett Nánási, Edina Katalin Cseh, Cecília Rajda, Gábor Veres, Eleonóra Spekker, Ágnes Szabó, Péter Klivényi, Masaru Tanaka, László Vécsei

**Affiliations:** 1Department of Neurology, Albert Szent-Györgyi Medical School, University of Szeged, Semmelweis u. 6, H-6725 Szeged, Hungary; polyak.helga@med.u-szeged.hu (H.P.); cseh.edina@med.u-szeged.hu (E.K.C.); rajda.cecilia@med.u-szeged.hu (C.R.); szabo.agnes.4@med.u-szeged.hu (Á.S.); peter.klivenyi@med.u-szeged.hu (P.K.); 2Doctoral School of Clinical Medicine, University of Szeged, Korányi fasor 6, H-6720 Szeged, Hungary; 3Department of Pediatrics, Albert Szent-Györgyi Faculty of Medicine, University of Szeged, H-6725 Szeged, Hungary; galla.zsolt@med.u-szeged.hu; 4Danube Neuroscience Research Laboratory, ELKH-SZTE Neuroscience Research Group, Eötvös Loránd Research Network, University of Szeged (ELKH-SZTE), Tisza Lajos krt. 113, H-6725 Szeged, Hungary; nannik1026@gmail.com (N.N.); spekker.eleonora@gmail.com (E.S.); tanaka.masaru.1@med.u-szeged.hu (M.T.); 5Independent Researcher, H-6726 Szeged, Hungary; weresdzsi@gmail.com

**Keywords:** multiple sclerosis, PPMS, SPMS, tryptophan, kynurenine, cuprizone, demyelination, remyelination, animal model, translational

## Abstract

Progressive multiple sclerosis (MS) is a chronic disease with a unique pattern, which is histologically classified into the subpial type 3 lesions in the autopsy. The lesion is also homologous to that of cuprizone (CPZ) toxin-induced animal models of demyelination. Aberration of the tryptophan (TRP)-kynurenine (KYN) metabolic system has been observed in patients with MS; nevertheless, the KYN metabolite profile of progressive MS remains inconclusive. In this study, C57Bl/6J male mice were treated with 0.2% CPZ toxin for 5 weeks and then underwent 4 weeks of recovery. We measured the levels of serotonin, TRP, and KYN metabolites in the plasma and the brain samples of mice at weeks 1, 3, and 5 of demyelination, and at weeks 7 and 9 of remyelination periods by ultra-high-performance liquid chromatography with tandem mass spectrometry (UHPLC-MS/MS) after body weight measurement and immunohistochemical analysis to confirm the development of demyelination. The UHPLC-MS/MS measurements demonstrated a significant reduction of kynurenic acid, 3-hydoxykynurenine (3-HK), and xanthurenic acid in the plasma and a significant reduction of 3-HK, and anthranilic acid in the brain samples at week 5. Here, we show the profile of KYN metabolites in the CPZ-induced mouse model of demyelination. Thus, the KYN metabolite profile potentially serves as a biomarker of progressive MS and thus opens a new path toward planning personalized treatment, which is frequently obscured with immunologic components in MS deterioration.

## 1. Introduction

Multiple sclerosis (MS) is an inflammatory, neurodegenerative, and immune-mediated disorder of the central nervous system (CNS), which is characterized by the appearance of inflammatory lesions of the white matter, axonal damage, loss of oligodendrocyte and axons, gliosis, demyelination, as well as neurodegeneration [1,2,3,4,5]. MS is still an incurable disease and is estimated to affect 2.8 million people worldwide [6]. It occurs most often in one of the most productive stages of life, i.e., in young adulthood, which makes the disease to be responsible for worsening the quality of the life not only for those affected but for society too [6,7]. The development of the disorder is the result of an interrelationship of various immune, genetic, epigenetic, and environmental factors [5]. The pathophysiology of the disease and the molecular and metabolic mechanisms underlying neuroaxonal damage are not yet fully elucidated, although oxidative stress greatly contributes to the progression of MS by inducing axonal and neuronal damage [5,8]. Furthermore, a growing body of research suggests that in addition to immune-mediated inflammatory responses, certain neurodegenerative processes, including glutamate excitotoxicity, mitochondrial dysfunction, and aforementioned oxidative stress also play a role in the pathogenesis and progression of the disease [8]. In most cases, MS starts with a relapsing–remitting (RRMS) course, often followed over the years by a relapse-independent deterioration in neurological function, known as the progressive phase of MS (SPMS) [9,10]. Progressive MS relates to gradual exacerbation in neurologic as well as psychiatric signs and symptoms. Cortical demyelinated lesions may be the pathological course leading to neural dysfunction, progression, and particularly cognitive impairments [11,12,13,14,15,16,17], which are clinical findings in patients with progressive MS. Extensive cortical demyelination has been observed in the frontal, temporal, insular, and cerebellar cortices, the cingulate gyrus, and the hippocampus [18,19].

Preclinical research makes a major contribution to prevention, diagnosis, and therapeutic management in clinical medicine, employing in vitro systems and animal models of diseases [20,21,22,23,24]. Experimental research collecting the data from animal studies and animal models of neurodegenerative and neuropsychiatric diseases has provided valuable clues to understanding the roles of endogenous peptides, hormones, and metabolites, searching for useful biomarkers and exploring novel targets for the treatment of diseases [25,26,27,28,29]. Thus, extensive preclinical, clinical, and computational studies are under extended research for their potential translationability and synthesizability in the search for novel preventive, diagnostic, and interventional measures for neurodegenerative diseases [30,31,32,33,34].

Various animal models, such as the experimental autoimmune encephalomyelitis, Theiler’s murine encephalomyelitis virus, and cuprizone (CPZ) toxin model, have been applied for the investigation of the pathomechanism of MS [35,36]. The CPZ-induced model is an ideal tool to study the demyelination phase, concomitant pathological changes and processes, as well as remyelination, in the absence of a peripheral immune response and blood–brain barrier (BBB) disruption [35,36]. CPZ is a copper chelating agent that induces apoptosis in mature oligodendrocytes [4]. The chelating function may have a negative effect on copper cofactor-dependent mitochondrial enzymes in the respiratory chain [3]. Although the exact mechanism of the intoxication is not yet fully explained, it appears that mitochondrial dysfunction causes oxidative stress, which contributes to the apoptosis of oligodendrocytes and axonal damage, ultimately leading to demyelination [3,4,37]. During the early weeks of toxin treatment (acute phase), the activation of oligodendrocytosis, microglia, and astrocyte starts, and demyelination becomes apparent [36,38,39]. Glial activation facilities the removal of accumulating myelin debris and gliosis further exacerbates demyelination and oligodendrocytosis [36,38,40,41]. Significant demyelination and oligodendroglial cell death can be observed in the corpus callosum, the striatum, the cortex, the hippocampus, and the cerebellum, among others [36]. Based on the literature, the weight of the brain decreases after CPZ treatment, which reduction can be explained by the thinning of the corpus callosum and cortex [35]. Furthermore, the discontinuation of CPZ treatment leads to rapid regeneration, exemplified by remyelination [35,42,43].

The CPZ intoxication contributes to axonal and synaptic damage, leading to the development of functional impairment of the nervous system [36,44]. Furthermore, CPZ intoxication causes glutamate excitotoxicity and synaptotoxicity through the glutamate receptor subunits [36,45,46]. The ionotropic amino-3-hydroxy-5-methyl-4-isoxazolepropionic acid (AMPA) and kainate types of glutamate receptors, and *N*-methyl-*D*-aspartate (NMDA) receptors in the myelin sheaths expressed in the oligodendrocytes, contribute to the cell and myelin sheath swelling, vacuolation, and subsequent excitotoxic cell death [35,47]. CPZ treatment also causes decreased expression of AMPA receptors in hippocampal neurons and cortex [46], upregulation of the NR2A subunit of the NMDA receptor in the corpus callosum [45], and downregulation of the mGluR2 subunit of the metabotropic glutamate receptors [36,45]. 

More than 95% of the essential amino acid tryptophan (TRP) is degraded via the kynurenine (KYN) metabolic pathway (Figure 1) [48,49]. The first rate-determining step of the TRP-KYN metabolism is the conversion of TRP to L-KYN, catalyzed by TRP-2,3-dioxygenase (TDO) and indoleamine-2,3-dioxygenases (IDOs) [50,51]. TDO can be found in liver cells and, to a lesser extent in the brain, while the IDOs are expressed in the neurons, the astrocytes, and the microglia [51]. The KYN metabolites are formed via different subbranches. Kynurenine aminotransferases (KATs) catalyze the transamination of KYN to form KYNA, whereas kynureninase catalyzes KYN to form ANA. Those metabolites are further degraded to form 3-hydroxy-L-kynurenine (3-HK), xanthurenic acid (XA), 3-hydroxyanthranilic acid (3-HANA), picolinic acid (PICA), and quinolinic acid (QUIN) [8,50,52,53]. During TRP degradation, many KYN metabolites were shown to possess neuroactive properties, such as KYNA and PICA, whereas QUIN and 3-HANA were considered to have neurotoxic properties [52,54]. KYNA affects glutamatergic transmission at various glutamate receptor subunits, including NMDA, AMPA, or kainate receptors [55,56,57]. A certain concentration of KYNA has been considered neuroprotective, scavenging the insult of reactive oxygen species [48,58]. On the other hand, QUIN is a competitive agonist of the NMDA receptors, and it may cause glutamatergic excitotoxicity by Ca^2+^ influx, which produces neuronal cell death [52,59,60].

Moreover, several studies have reported the alteration of TRP-KYN metabolites in neurodegenerative diseases and psychiatric disorders, including MS [48,52,61,62,63,64]. It was reported that KYNA levels were elevated in the relapse phase while decreased in the remission phase [65,66]. Meanwhile, reduced KYNA and PICA and elevated QUIN concentrations were reported in the cerebrospinal fluid of MS patients [48]. In addition, decreased KYNA levels were observed in the progressive phase of the disease [67]. A recent study has reported a decreased 3-HK level in the serum samples of MS patients [68].

Recently, we reported a significant decrease in KYNA levels in the cortex and the hippocampus, and the plasma during the demyelination period, which is normalized after remyelination in the CPZ-induced model of demyelination [69]. In this study, we measure the levels of serotonin, TRP, and eight TRP-KYN metabolites of the plasma and the brain samples of CPZ-intoxicated mice in the demyelination and remyelination periods in search of a TRP metabolite profile of progressive and recovery phases.

## 2. Materials and Methods

### 2.1. Animal Experiments and Sample Collection

In the experiment, eight-week-old C57BL/6J male mice were used (*n* = 160). The animals were bred and maintained under standard laboratory conditions with 12 h–12 h light/dark cycle at 24 ± 1 °C and 45–55% relative humidity in the Animal House of the Department of Neurology, University of Szeged. The investigations were in accordance with the Ethical Codex of Animal Experiments and were approved by the Ethics Committee of the Faculty of Medicine, University of Szeged, and the National Food Chain Safety Office with a permission number of XI/1101/2018. The experiment was performed as previously described in our study [69]. Briefly, the animals were housed in polycarbonate cages (530 cm^3^ floor space) in groups of 5. Prior to the start of the experiment, all animals were acclimated to grounded standard rodent chow for 2 weeks, and animal’s weight was monitored every other day.

The CPZ toxin was administered to half of the experimental animals (*n* = 80) for 5 weeks by a diet containing 0.2% CPZ (bis-cyclohexanone-oxaldihydrazone; Sigma-Aldrich) mixed into a grounded standard rodent chow with free access to water. For control group (CO), age and weight-matched animals were used (*n* = 80), which had rodent chow and free access to water. At the end of the first, third, and fifth weeks, 16 animals were randomly chosen from both CO and CPZ groups and terminated for further analysis. Thus, at the end of the demyelination phase, 96 animals were terminated (*n* = 96, 48 CPZ-treated, and 48 control animals). The surviving animals (*n* = 64, 32 CPZ treated, and 32 control animals) underwent the remyelination phase for 4 weeks and they were sacrificed at the end of the second and fourth weeks of the recovery phase (Figure 2).

The animals were terminated according to our previous investigation [69]. The mice were anesthetized with intraperitoneal 4% chloral hydrate (10 mL/kg body weight). For the histological and immunohistochemical studies, mice (CPZ: *n* = 30, CO: *n* = 30) were perfused transcardially with artificial cerebrospinal fluid followed by 4% paraformaldehyde in 0.1 M phosphate buffer. Brain samples were dissected and postfixed in the same fixative overnight at 4 °C. Brains were embedded in paraffin, coronally sectioned in 8 μm thickness obtained from different regions (0.14, −0.22, −1.06, and −1.94 mm) according to the mouse brain atlas of Paxinos and Franklin (2001) and placed on gelatin-coated slides [70]. For bioanalytical measurements, the animals (CPZ: *n* = 50, CO: *n* = 50) were anesthetized and perfused as described above. Blood samples were taken from the left heart ventricle into Eppendorf tubes containing 5% disodium ethylenediaminetetraacetate dihydrate and plasma was separated by centrifugation (3500 rpm for 10 min at 4 °C). The brains were dissected into five different brain regions: the cerebellum, the brainstem, the striatum, the somatosensory cortex, and the hippocampus. All samples were placed on ice and stored at −80 °C. The samples were marked as groups of DEM, REM, or CO.

### 2.2. Luxol Fast Blue Crystal Violet Staining and Myelin Status Determination by Densitometric Analysis

Myelin damage was examined by luxol fast blue crystal violet staining. The brain slides were deparaffinized, rehydrated with 95% alcohol, and incubated in a 0.01% luxol fast blue solution overnight at 60 °C, after that the sections were differentiated in 0.05% lithium carbonate solution and counterstained with crystal violet. For measurements of the stained sections were taken using a Zeiss AxioImager M2 microscope, supplied with an AxioCam MRc Rev. 3 camera (Carl Zeiss Microscopy). Zeiss Zen 2.6 (blue edition)^®^ image analysis software program was applied, which measured the mean intensity of different color channels on a scale from 0 to 65,536. In our case, the low-intensity value characterizes the higher myelin content in the control (CO) group as there was a higher tissue staining. On the other hand, the higher intensity measured in the CPZ-treated (DEM) group shows a lower rate of tissue staining resulting in a decreased myelin content. To determine the myelin content, we performed luxol fast blue crystal violet staining, then the corpus callosum was marked on each section based on the mouse brain atlas of Paxinos and Franklin (2001), and intensity measurement was used on this designated area in order to determine myelin content in the corpus callosum.

### 2.3. Ultra-High-Performance Liquid Chromatography with Tandem Mass Spectrometry Measurement

All reagents and chemicals were of analytical or liquid chromatography–mass spectrometry grade. TRP and its metabolites, and their deuterated forms: d4-SERO, d5-TRP, d4-KYN, d5-KYNA, d4-XA, d5-5-HIAA, d3-3-HANA, d4-PICA, and d3-QUIN were purchased from Toronto Research Chemicals (Toronto, ON, Canada). d3-3-HK was obtained from Buchem B. V. (Apeldoorn, The Netherlands). Acetonitrile (ACN) was provided by Molar Chemicals (Halásztelek, Hungary). Methanol (MeOH) was purchased from LGC Standards (Wesel, Germany). Formic acid (FA) and water were obtained from VWR Chemicals (Monroeville, PA, USA).

The preparation of the standards, internal standards (IS), and quality control (QC) solutions were necessary for the measurement, as well as the preparation of the animal plasma sample for analysis, which was based on the description published by Tömösi et al. [71]. As for the brain samples, after measuring the weight of the five different brain regions, we homogenized them (UP100H, Hielscher Ultrasound Technology, Germany; amplitude: 100%, cycle: 0.5) in 3× amount of ice-cooled LC-MS water (for example, 90 μL water was pipetted to 30.0 mg sample). After that, the same steps as for the plasma samples were performed, with the difference that the precipitation was carried out with 100% acetonitrile. Then, plasma samples and brain regions were measured according to the previously published methodology using UHPLC-MS/MS [72]. Multiple reaction monitoring (MRM) transition of picolinic acid was 124.0/106.0 using 75 V as declustering potential and 13 V as collision energy, retention time: 1.21 min.

### 2.4. Statistical Analysis

For the statistical analysis of body weight, two-way repeated-measures ANOVA was used. For the densitometric analysis of LFB statistical differences were determined by one-way analysis of variance (ANOVA), then depending on the variances of data, Sidak or Tamhane’s T2 post hoc test was applied. Pairwise comparisons of group means were based on the estimated marginal means with Sidak or Tamhane’s T2 post hoc test with adjustment for multiple comparisons. Group values were given as means ± SEM, analyses were performed in SPSS Statistics software (version 20.0 for Windows, SPSS Inc. IBM, Armork, NY, USA). Regarding the UHPLC-MS/MS measurements, all statistical analyses were performed with the help of the R software (R Development Core Team). After checking for its assumptions (checking for outliers, Shapiro and Levene tests), we performed two-way ANOVA with estimated marginal means post hoc tests to determine significance between treatment groups, measurement times, and their interaction. In case of the assumptions were not met, we applied the Sheirer–Ray–Hare test with Dunn test as post hoc. Type I errors from multiple comparisons were controlled with the Bonferroni method. We rejected null hypotheses when the corrected *p* level was < 0.05, and in such cases, the differences were considered significant.

## 3. Results

### 3.1. Investigation of Body Weight

The results of body weight measurement during the CPZ treatment and the recovery phase can be seen in the Appendix A.

### 3.2. Evaluation of Cuprizone Damage in the Demyelination and Remyelination Phases

The extent of myelin damage was examined by luxol fast blue crystal violet staining. A detailed description and figures of the immunohistochemical analyses and subsequent intensity measurements can be found in the Appendix A.

### 3.3. Ultra-High-Performance Liquid Chromatography with Tandem Mass Spectrometry Measurement of Kynurenine Metabolites 

During our experiment, UHPLC-MS/MS bioanalytical measurements were performed to detect TRP and different KYN metabolites from both plasma and different brain region samples, including the striatum, cortex, hippocampus, cerebellum, and brainstem. These samples were collected on the 1st, 3rd, and 5th week of treatment, as well as on the 2nd and 4th week of the recovery phase. Due to the large amount of data obtained, only the significant changes were shown and published. Namely, in the case of plasma, we observed a significant difference in the levels of KYNA, 3-HK, and XA in terms of the CPZ treatment time and the degree of damage. Already, at the beginning of the CPZ treatment, a significant decrease in the level of these metabolites was observed, and differences persisted until the end of treatment. In the first half of the recovery phase, these concentration differences disappeared while the level of KYNA, 3-HK, and XA metabolites was in the same range in both groups until the end of remyelination (detailed in Figure 3). In addition to the mentioned metabolites, at the 5th week of treatment, there was also a difference in plasma TRP and ANA levels between CO and CPZ groups (Figure 4).

In the analysis of the brain regions, a significant difference was detected in the concentration of 3-HK, already in the first week of the CPZ treatment in the striatum. However, in the 3rd week of the CPZ intoxication, in addition to the striatum, a significant reduction in 3-HK concentration was also noticed in the cortex, the hippocampus, and the brainstem, which was still maintained in the 5th week of CPZ treatment and was even more pronounced in the cortex and the hippocampus (Figure 5).

In addition, in the 3rd week of treatment, an increase in TRP concentration was observed in the cortex and hippocampus in the CPZ group. The elevated TRP level became even more apparent by the 5th week of CPZ intoxication, and it was also increased in the striatum and likewise in the cortex and the hippocampus (Figure 6).

Moreover, in addition to the differences in 3-HK and TRP levels, there was also a significant decrease in ANA concentration by the 5th week of CPZ intoxication in the striatum, the cortex, the hippocampus, and the brainstem of the CPZ group compared to the CO (Figure 7).

## 4. Discussion

In this study, we examined the TRP-KYN metabolites in the CPZ-induced animal model of demyelination. The CPZ-treated model is a widely used model of MS, which is considered to be spared the participation of immunological components. Histologically, the model is characterized by oligodendrocyte apoptosis, demyelination, as well as microglia and macrophage activation in the absence of inflammatory processes, which provides an opportunity to investigate the progressive phase of MS in the preclinical model. To ensure the reliability of the CPZ-induced model, we monitored the changes in the body weight of the animals, and then we performed immunohistochemical analyses to determine the degree of demyelination in the corpus callosum. 

Bioanalytical measurements were applied to investigate serotonin (5-HT), TRP, and KYN metabolites in the plasma and various brain regions, including areas affected by demyelination and oligodendrocytosis-induced damage [35,36]. This complex bioanalytical study was made to expand our knowledge obtained in the previous study regarding the measurement of four KP metabolites in the CPZ-induced model of MS [69].

In a series of our studies, we noticed a significant body weight decrease in the CPZ-treated animals a few days after the CPZ treatment. Upon the withdrawal of the CPZ toxin, the difference in the body weight starts decreasing and the CPZ-treated mice gained as much weight as that of the CO group by the end of the remyelination phase. Histologically, we observed myelin damage at the beginning of the CPZ treatment, which becomes more significant over time and progressed to extensive severe demyelination. These changes are in line with the previous data [35,36].

Collecting samples at a different time frames made it possible to monitor the progress and extent of the damage, as well as to analyze changes in the concentrations of various metabolites. At the end of the first week of CPZ treatment, immunohistochemical analyzes did not show any difference in myelin sheath between the CPZ and CO groups, as presumably, demyelination damage was not detectable [35,36,73]. By the end of the third week, significant myelin damages were observed in the CPZ toxin-treated group, including oligodendrocyte degeneration, and significant microglia and astrocyte activation processes were observed in this phase [36,73]. By the fifth week, we observed large and severe demyelination in the corpus callosum of the CZP group. The significant demyelination, the phagocytosis of the myelin sheath by microglia, extensive and severe axon damage, astrocytosis, and microgliosis are also supported by the literature data [35,36,73].

These observations have encouraged us to explore a metabolic change in TRP metabolism in response to CPZ treatment, particularly during the demyelination and remyelination phases. Thus, UHPLC-MS/MS analysis was applied to measure the concentrations of 5-HT, TRP, and KYN metabolites. In the plasma samples, reduced levels of KYNA, XA, and 3-HK were measured as early as in the first week in the CPZ-treated group. The difference persisted toward the third and fifth weeks of CPZ treatment. Furthermore, by the fifth week, the differences in both TRP and ANA levels were observed in the CPZ-treated group. During the remyelination phase, the differences in those metabolites decrease to become insignificant and become none eventually.

In this study, by performing immunohistochemical analyses and plasma metabolite concentration measurements performed at different times of CPZ treatment and recovery phase, we wanted to investigate whether the concentration of KYN metabolites shows the differences already at the beginning of the intoxication and how the levels change during the CPZ treatment, whether they are consistent with the demyelination and remyelination processes in the periphery and the central nervous system. At the beginning of CPZ poisoning, oligodendrocytosis starts already, resulting in demyelination, and later micro-glia and macrophage activation occurs (for more details, see [36]). During the plasma examination, already in the first week of the CPZ treatment, we observed significant concentration differences for certain metabolites between the groups, which became even more evident by the 5th week, but by the 2nd week of recovery, the differences has disappeared, and it seems that the levels of KYN metabolites had normalized by remyelination process.

However, CPZ treatment alters normal liver function due to the megamitochodrium formation, and as a result, plasma amino acid levels also change during treatment [74]. In addition to our immunohistochemical analyses, we found differences in both weight and TRP-KYN metabolite levels as a result of CPZ treatment in the demyelination phase. However, in the recovery phase, these differences between the groups disappeared and it seems that parallel to body weights, metabolite concentrations also normalized relatively quickly in the remyelination process.

Similarly, we observed a significant decrease in 3-HK levels in the brain samples including the striatum, the cortex, the hippocampus, and the brainstem, and by the end of the fifth week, a marked decrease in both 3-HK and ANA levels was observed in the same brain regions. Interestingly, the TRP levels were elevated during CPZ treatment in the cortex and the hippocampus and by the fifth week in the striatum. The differences likewise disappeared in the remyelination phase upon CPZ withdrawal, suggesting the remedy of demyelination. Moreover, in the present study, we observed a significant 3-HK reduction both in the periphery and the brain tissues. Saraste and colleagues reported a reduced level of 3-HK in patients with MS, which is considered to be related to microglial activity [68]. Similarly, a significant microglia/macrophage activation has been described already in the first few weeks of treatment, which persists up to five weeks during the demyelination period in the CPZ animal model [35,36]. The dysregulation of TRP-KYN metabolism is significantly associated with neurodegenerative processes, especially via microglia activation [75,76,77]. Furthermore, the correlation between microglial activation and low 3-HK levels may be ascribed to the genetic variability of the enzymes, the locational difference in enzyme activity in the body, and the resultant difference in metabolite levels, employed in TRP-KYN metabolism [68,78]. The BBB also plays an important role in a locational difference in the metabolite concentrations: 3-HK and TRP cross the BBB efficiently, while KYNA and QUIN hardly cross it, while the BBB remains relatively intact during CPZ treatment [68,79,80]. Additionally, a reduced 3-HK plasma level was also observed in patients with major depressive disorder compared to the healthy controls [81]. A significant decrease in 3-HK levels was observed in the serum of schizophrenic patients [82]. Thus, these diseases may be associated with microglial activation [83,84,85]. 

Some KYN metabolites are neuroactive. Furthermore, the changes in the level of KYN metabolites can affect the activity of enzymes. CPZ intoxication may affect the enzyme functions in KYN metabolism. Indeed, it was shown that elevated copper concentration affects the function of the KAT enzymes in the periphery [86] and presumably thereby the level of KYNA. Probably, this may explain the decrease in KYNA levels in the periphery, as a result of CPZ treatment.

Based on studies, the copper–zinc superoxide dismutase cuproenzyme also shows reduced activity when treated with CPZ [87,88,89]. Copper, as a cofactor of various copper enzymes, among others superoxide dismutase [90], dopamine-β-hydroxylase [91], monoamine oxidase [92], the cytochrome c oxidase family [93], cytochrome c oxidase assembly protein [94,95]; plays a significant role in several cellular processes, and neurodegeneration may develop in the event of a disturbance in its homeostasis [35].

In contrast, 3-HK and QUIN are considered neurotoxic metabolites at least in certain concentrations and environments, inducing cell death by various excitotoxic processes [75,96,97]. Recently, an increasing amount of evidence highlighted that 3-HK is not always toxic, because 3-HK together with 3-HANA may contribute to the regulation of the redox balance of brain tissue and prevent further damage [75,98,99]. Therefore, 3-HK is a controversial metabolite, as it can function as a scavenger but promote oxidative damage, depending on the redox environment, pH, or cell type, among others [99]. As for XA, studies suggest that XA has antioxidant properties, as it can inhibit lipid peroxidation and prevent iron-induced NADP-isocitrate dehydrogenase inactivation [99,100,101]; furthermore, it can scavenge free radicals [99,102], bind the superoxide anion, and inhibit hematoxylin autoxidation (see the review [99]).

ANA has been attracting increasing attention as a neuroprotective agent. ANA has proven to be a strong radical scavenger by effectively binding the largest subset of free radicals [99,102]. Moreover, ANA affects respiratory parameters [103] and is a precursor of the synthesis of nonsteroidal anti-inflammatory drugs, [63,104]. Considering this, XA and ANA may play a major role as antioxidants in CPZ intoxication. Presumably, the elevated TRP levels of the CNS can be explained by the reduction of enzyme activities of downward KYN enzymes and/or a compensatory mechanism to ensure TRP availability due to CPZ intoxication. In a recent study, a significantly higher TRP level was observed among patients with MS compared to the control group [105].

The metabolite concentration differences observed in individual brain regions may be related to changes in the volume of certain regions.

CPZ-induced oligodendrocytosis is unequally distributed in the CNS. CPZ poisoning causes extensive oligodendrocytosis and severe demyelination, among others in the corpus callosum, cerebral cortex, hippocampus, and to a lesser extent in the cerebellum and brainstem. The reasons for the regional variability are not known, but it may be influenced by the uneven distribution of different oligodendrocyte subtypes in the CNS. Thus, it may happen that CPZ is much more toxic to some subtypes, while less to others. In addition, altered gene expression can affect the sensitivity of certain areas to injuries (for more details, see [36]).

Furthermore, based on studies, the dry mass of the brain irreversibly reduced after CPZ intoxication [106,107], which was also indicated by the thinning of the corpus callosum and cortex after CPZ treatment [35,108,109,110]. In our study, we found a significant difference in metabolite concentration in the brain regions, that were mentioned in the literature as severely demyelinated areas, including the cortex and hippocampus as well as the brainstem.

Moreover, based on the literature data, CPZ also causes damage to neurotransmitter homeostasis. CPZ exerts an inhibitory effect on glutamic acid decarboxylase, an increase in glutamate (GLU) level, and a decrease in the gamma-aminobutyric acid (GABA) [111]. Another study, on the other hand, described an increased GABA level during 3 weeks of CPZ treatment, in contrast to the reduced GABA level seen during 8 weeks of treatment, which may point to changes in neurotransmitter concentration over time during CPZ intoxication [35,112]. Furthermore, based on research, dopaminergic and noradrenergic synapses are also affected during CPZ treatment. Specifically, CPZ poisoning has an inhibitory effect on the functioning of dopamine hydroxylase and monoamine oxidase enzymes, which affect dopamine and norepinephrine concentrations (for more details, see [35]).

The authors acknowledge the limitations of this study and different outcomes depending on analytical methods. In the present study, UHPLC-MS/MS analysis did not re-produce the TRP and KYNA concentrations in the brain regions, which were measured by the HPLC method. The discrepancy may be ascribed to the difference in sample preparation (precipitation with perchloric acid vs. acetonitrile) and in detection methods (UV–VIS vs. UHPLC-MS/MS). Interestingly, the UHPLC-MS/MS method measures KYNA in a much narrower concentration range from brain samples, compared to measurements with fluorescent detectors [113]. Recently, more bioanalytical measurements were carried out by mass spectrometry analysis in the advantage of its outstanding selectivity, sensitivity, detection specificity, and reproducibility, compared to HPLC methods.

Many factors play a role in the examination and analysis of brain regions, such as the complexity of the regions, sample preparation and its limitations, and many additional different metabolites present in the CNS. Nevertheless, it is important to mention a problem that affects many and is becoming increasingly common, namely the reproducibility crisis including using different analytical methods. The scientists have drawn attention to, and focus on the dilemma that the repetition of studies is not necessarily reproducible in animal as well as in human studies [114].

Nonetheless, this is the first study comprehensively showing the concentrations of 5-HT, TRP, and KYN metabolites at different time points during CPZ treatment and the recovery phase, in the plasma and five different brain regions. The study has confirmed metabolic changes on both sides of the BBB during demyelination. In accordance with our previous study, here we showed the involvement of KYN metabolites in the CPZ-induced animal model of demyelination, which is analog to progressive MS in terms of some markers. In addition to complementing the previous data, this study has revealed the profile of KYN metabolites during progressive demyelination. Further studies might shed light on the mechanism behind the alteration of KYN metabolites and possible changes in enzyme activities of TRP-KYN metabolism.

## Figures and Tables

**Figure 1 biomedicines-11-00945-f001:**
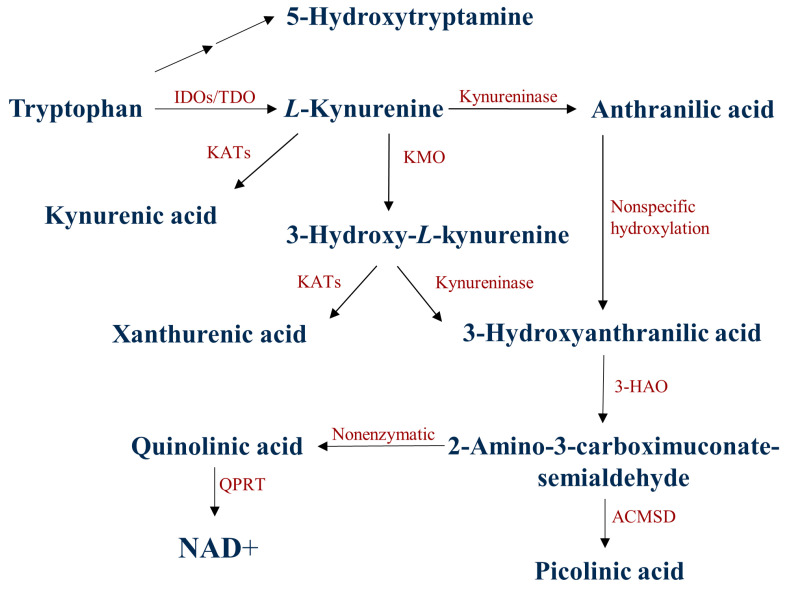
The kynurenine metabolic pathway of tryptophan degradation. 3-HAO: 3-hydroxyanthranilate oxidase; ACMSD: α-amino-β-carboxymuconate-semialdehyde-decarboxylase; IDOs: indoleamine 2,3-dioxygenases; TDO:/tryptophan 2,3-dioxygenase; KATs: kynurenine aminotransferases; KMO: kynurenine 3-monooxygenase; NAD^+^: nicotinamide adenine dinucleotide: QPRT: quinolinate phosphoribosyltransferase.

**Figure 2 biomedicines-11-00945-f002:**
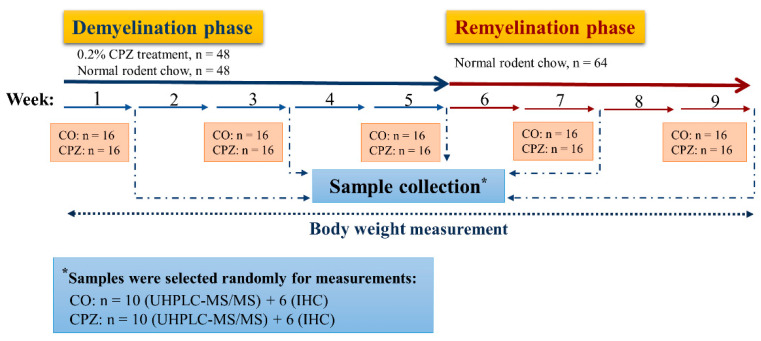
Timeline of the experimental procedure used. CO: control group; CPZ: cuprizone group; IHC: immunohistochemical studies; n: the number of animals; UHPLC-MS/MS: Ultra-high-performance liquid chromatography with tandem mass spectrometry; *: random sample selection was applied for both measurements in the CPZ-treated and the CO groups.

**Figure 3 biomedicines-11-00945-f003:**
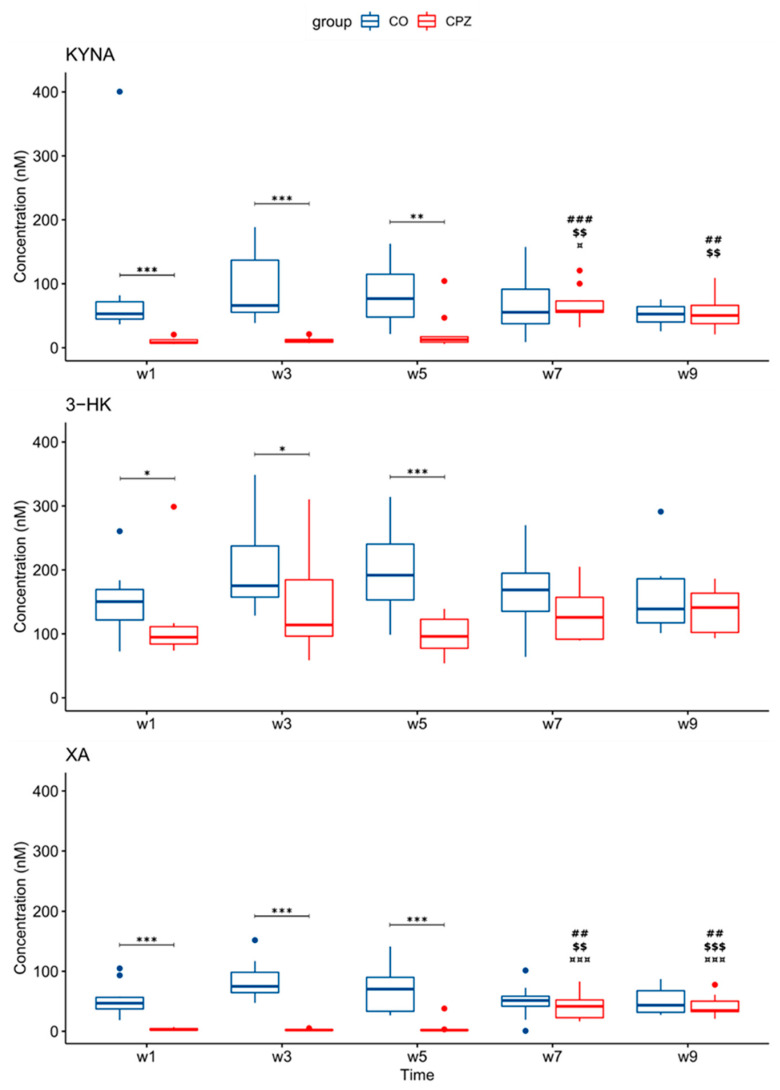
Changes in kynurenine metabolites during weeks of cuprizone treatment in the plasma of cuprizone-treated and control groups. In the first week of the CPZ treatment, a significant decrease in KYNA, 3-HK, and XA levels was observed in the CPZ group compared to the control, and differences remained until the end of the CPZ intoxication. In the first half of the recovery phase, these differences disappeared during remyelination between groups. 3-HK: 3-hydroxy-L-kynurenine; CO: control group; CPZ: cuprizone group; KYNA: kynurenic acid; XA: xanthurenic acid; w: week; *: *p* < 0.05: **: *p* < 0.01; ***: *p* < 0.001: ##: *p* < 0.01; ###: *p* < 0.001 compared to week 1; $$: *p* < 0.01; $$$: *p* < 0.001 compared to week 3; ¤: *p* < 0.05; ¤¤¤: *p* < 0.001 compared to week 5.

**Figure 4 biomedicines-11-00945-f004:**
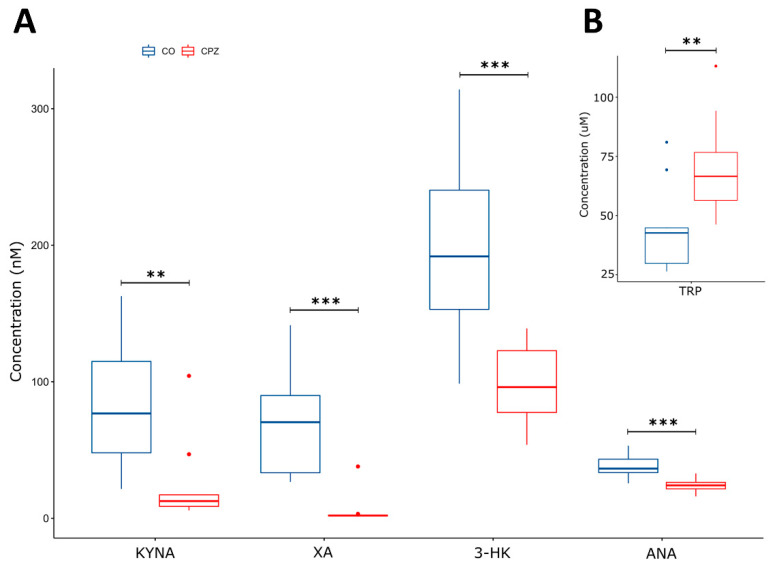
Alteration in plasma metabolite levels at week 5 of CPZ treatment. In addition to the KYNA, 3-HK, and XA (nM), we also observed a significant difference in the concentration of ANA (nM) (**A**) and TRP (µM) metabolites (**B**) by the 5th week of CPZ treatment between CO and CPZ groups. CO: control group; CPZ: cuprizone group: 3-HK: 3-hydroxy-L-kynurenine; ANA: anthranilic acid; KYNA: kynurenic acid; TRP: tryptophan; XA: xanthurenic acid; w: week; **: *p* < 0.01; ***: *p* < 0.001.

**Figure 5 biomedicines-11-00945-f005:**
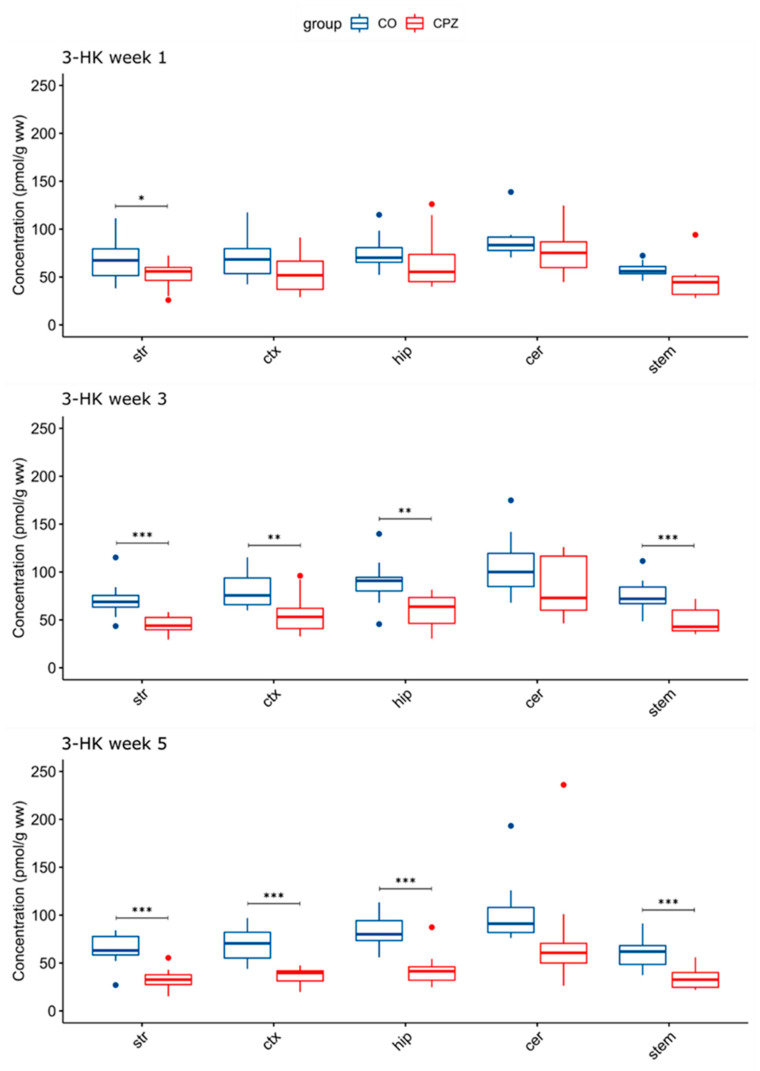
Decreased 3-hydroxy-L-kynurenine concentrations in certain brain regions in time progression in the cuprizone-treated group. A significant difference in 3-HK levels emerged between the groups as the damage caused by the CPZ treatment worsened. 3-HK: 3-hydroxy-L-kynurenine; CO: control group; cer: cerebellum: CPZ: cuprizone group: ctx: cortex; hip: hippocampus; stem: brainstem; str: striatum; ww: wet weight; *: *p* < 0.05; **: *p* < 0.01; ***: *p* < 0.001.

**Figure 6 biomedicines-11-00945-f006:**
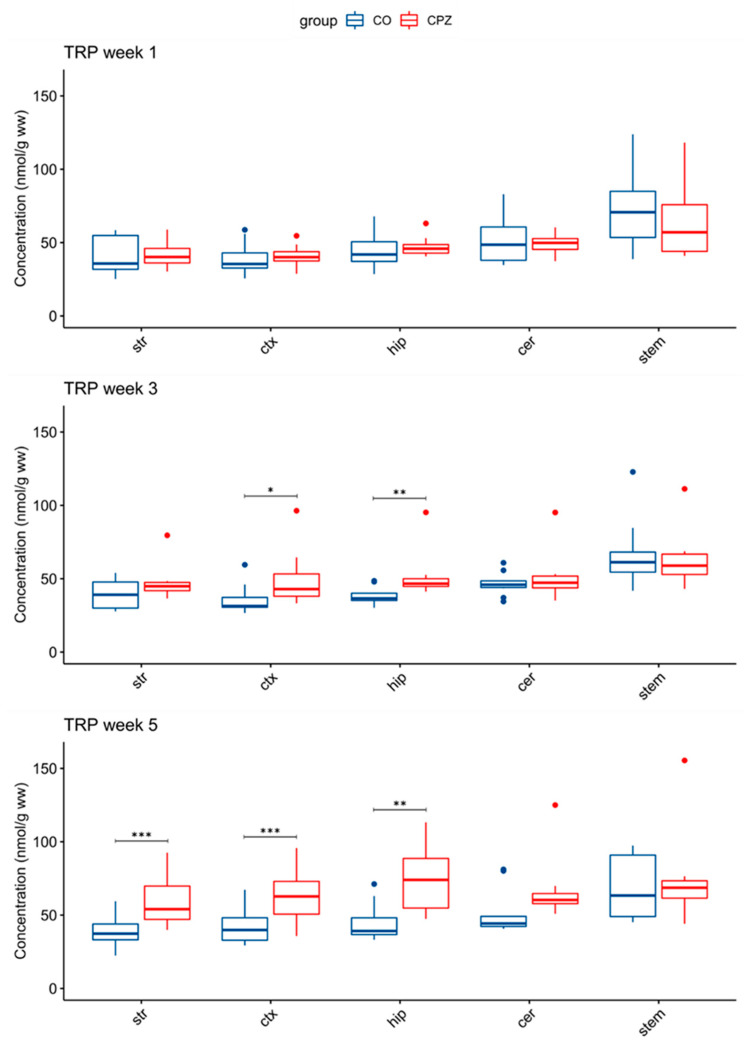
Elevated tryptophan concentrations in some brain regions in the cuprizone-treated group. In the 3rd week of CPZ treatment, the TRP levels were significantly increased in the cortex and the hippocampus. By the 5th week, the TRP concentration was also significantly increased in the striatum during the CPZ treatment. cer: cerebellum; CO: control group; CPZ: cuprizone group; ctx: cortex: hip: hippocampus; stem: brainstem: str: striatum; ww: wet weight; TRP tryptophan; *: *p* < 0.05; **: *p* < 0.01; ***: *p* < 0.001.

**Figure 7 biomedicines-11-00945-f007:**
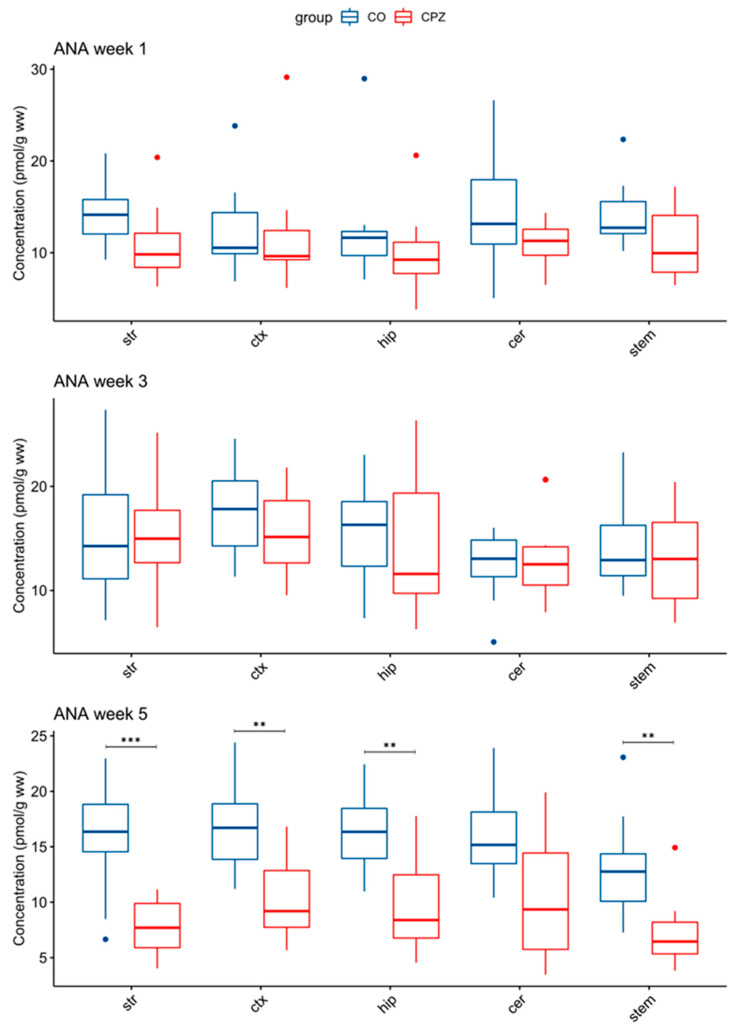
Change in concentration of ANA during cuprizone treatment. In the 5th week of CPZ treatment, a marked decrease in ANA concentrations in the striatum, the cortex, the hippocampus, and the brainstem in CPZ-treated group was observed, compared to the control group. ANA: anthranilic acid; cer: cerebellum; CO: control group; CPZ: cuprizone group; ctx: cortex; hip: hippocampus; stem: brainstem; str: striatum; ww: wet weight; **: *p* < 0.01; ***: *p* < 0.001.

## Data Availability

The data presented in this study are available on request from the corresponding author.

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
