# Peer review of "The Tryptophan-Kynurenine Metabolic System Is Suppressed in Cuprizone-Induced Model of Demyelination Simulating Progressive Multiple Sclerosis"

_biomedicines, 2023, doi:10.3390/biomedicines11030945_

Round 1

Reviewer 1 Report

This manuscript by Polyak et al. analyze levels of TRP and its metabolites in the mouse cuprizone demyelination model.   

I have the following comments/questions:

First, while most of the manuscript can be understood, the quality of the English is average. Many syntax and grammatical errors are found throughout the text. The authors should pay specific attention to the meaning of “however”.

As mentioned in lines 133-135, the authors have already recently published on the subject similar results. The authors should mention if the levels of KYNA observed here were comparable as in their first publication.

Figure 3, 4 and 5 could be included in supplementary materials. It simply shows that the authors can correctly use the cuprizone model. They could also simply refer to their previous article. However, since the data is provided, I have to mention that the quality of the images in Fig. 4 are poor to average. Additionally, some images are darker than others. Also, in fig. 5, the mean intensity of LFB increases in CPZ treated animals? It should be the reverse. The same error seems to be found in the author's previous paper (fig 7. https://doi.org/10.1016/j.heliyon.2021.e06124), where the authors show an increase intensity for both LFB and MBP during the demyelination phase where it should be decreased. The authors, however, do mention in the text a decrease.

In Fig. 7 of the submitted manuscript, the authors show a significant increase in levels of TRP in the plasma. This was not found in their previous paper. The authors should explain this discrepancy. Similarly, the authors now show a significant increase in levels of TRP in week 5 (which is equivalent to their previous demyelination time point) in the CNS, while no differences were observed in their previous publication. In addition, the authors should comment has to why the absolute levels observed in their previous article is much lower than in this paper.

In the discussion (lines 349, 375 and 438), the authors mention they have investigated the levels of 5-HT. No data on this was provided in this manuscript. In addition, (lines 368-371) the authors write that they observed “the phagocytosis of the myelin sheath by microglia, extensive and severe axon damage, astrocytosis, and microgliosis”. This was not shown. Similarly, lines 390-392, the data was not provided. Please verify the wording used in the discussion to make sure that everything that is referenced from the literature does not appear to be from this manuscript.

Lines 404-408: Why are the neuroprotective properties of KYNA discussed here if there is no difference observed in the CNS? This is could belong in the introduction.

Lines 410-412: If CPZ treatment affects the “functions of KAT enzymes and thus, the level of KYNA”, why is there no differences observed in the CNS?

 Lines 431-437: why is this mentioned? What does it bring to the manuscript apart from reminding the readers that even the authors obtained different results when doing the same experiment (difference in TRP levels compared to previous publication)?

 Line 443 and line 33: CPZ-induced demyelination is a model. It is useful to study biological principles that might also be happening during progressive MS, but it’s not because we observed something in the model that it is the same in patients. The authors need to rephrase these two sentences to make it clear that CPZ-induced demyelination is not the same as progressive MS.

 Overall, in the discussion, the authors fail to explain or put in context their results. As mentioned in lines 428-430 and 445-447, the levels of enzymes present in the tissue and their enzymatic activity would be needed to draw any significant conclusion from the data provided. As such, I fail to see the novelty in the dataset presented. The authors have more a less repeated their previous experiment and included more time points and a few more metabolites. They show that there are quantitative differences without explaining what it means, and they fail to draw conclusions or point to similarities with what is observed in patients, as alluded at the end of the abstract.   

 In my opinion, as it stands, the manuscript doesn’t reach the originality threshold and doesn’t provide enough new data to warrant publication in biomedicines. 

Author Response

Reviewer #1:

This manuscript by Polyak et al. analyze levels of TRP and its metabolites in the mouse cuprizone demyelination model.

The authors greatly appreciate the detailed comments and suggestions of the Reviewer. We would like to answer your suggestions to the best of our knowledge.

  • First, while most of the manuscript can be understood, the quality of the English is average. Many syntax and grammatical errors are found throughout the text. The authors should pay specific attention to the meaning of “however”.

Thank you for your observation, we carefully reviewed the manuscript and tried to correct the mistakes, with speciel regard to the meaning of „however”.

We changed the Page 1, Lines 24-26 as follows: „Aberration of the tryptophan (TRP)-kynurenine (KYN) metabolic system has been observed in patients with MS; nevertheless, the KYN metabolite profile of progressive MS remains inconclusive.”

We changed the Pg 3, Lines 80-83 as follows: „The pathophysiology of the disease, and the molecular and metabolic mechanisms underlying neuroaxonal damage are not yet fully elucidated, although oxidative stress greatly contributes to the progression of the MS by inducing axonal and neuronal damage [5,8].”

We changed the Pg 4, Lines 166-167 as follows: „A recent study has reported a decreased 3-HK level in the serum samples of MS patients [68].”

We changed the Page 10, Lines 347-349 as follows: „In the first half of the recovery phase, these concentration differences disappeared while the level of KYNA, 3-HK and XA metabolites were in the same range in both groups until the end of remyelination (detailed in Figure 6.).”

We changed the Page 11, Lines 363-364 as follows: „In the first half of the recovery phase, these differences disappeared during remyelination between groups.”

  • As mentioned in lines 133-135, the authors have already recently published on the subject similar results. The authors should mention if the levels of KYNA observed here were comparable as in their first publication.

Thank you for your suggestions. We clarify this study is an extention of the previous study, measuring 10 metabolites in search of a profile for progressive phase, including serotonin, tryptophan, and kynurenines, as follows: Page 12, lines 170-173 as follows: „In this study, we measure the levels of serotonin, TRP, and eight TRP-KYN metabolites of the plasma and the brain samples of CPZ-intoxicated mice in the demyelination and remyelination periods in search of a TRP metabolite profile of progressive and recovery phases.”

Furthermore, we acknowledge the discrepancy of the results of this study and those of the previous one, especially with the brain samples, which may be due to different sample preparation and analytical methods. UHPLC-MS/MS analysis was conducted according to the guidelines of European Medicines Agency and the second measurements have validated the accuracy of the first measurements.

We clarified it in the discussion section as a limitation of this study as follows: Page 17, Lines 527-536 „The authors acknowledge the limitations of this study and different outcomes depending on analytical methods. In the present study, UHPLC-MS/MS analysis did not reproduce the TRP and KYNA concentrations in the brain regions, which was measured by HPLC method. The discrepancy may be ascribed to difference in sample preparation (precipitation with perchloric acid vs. acetonitrile) and in detection methods (UV-VIS vs. UHPLC-MS/MS). Interestingly, the UHPLC-MS/MS method measures KYNA in a much narrower concentration range from brain samples, compared to measurements with fluorescent detectors [96]. Recently, more bioanalytical measurements were carried out by mass spectrometry analysis in advantage of its outstanding selectivity, sensitivity, detection specificity, and reproducibility, compared to HPLC methods.”

  • Figure 3, 4 and 5 could be included in supplementary materials. It simply shows that the authors can correctly use the cuprizone model. They could also simply refer to their previous article. However, since the data is provided, I have to mention that the quality of the images in Fig. 4 are poor to average. Additionally, some images are darker than others. Also, in fig. 5, the mean intensity of LFB increases in CPZ treated animals? It should be the reverse. The same error seems to be found in the author's previous paper (fig 7. https://doi.org/10.1016/j.heliyon.2021.e06124), where the authors show an increase intensity for both LFB and MBP during the demyelination phase where it should be decreased. The authors, however, do mention in the text a decrease.

Thank you for your observation. Indeed, Figures 3., 4., and 5. support the proper functioning of the cuprizone model. We also wanted to prove that the body weight change and immunohistochemical analyzes used in our study confirm that cuprizone toxin-induced animal modell works properly, and we wanted to reproduce the immunohistochemical changes already described in the literature during the different weeks of treatment (Praet J, Guglielmetti C, Berneman Z, Van der Linden A, Ponsaerts P. Cellular and molecular neuropathology of the cuprizone mouse model: clinical relevance for multiple sclerosis. Neurosci Biobehav Rev 2014;47:485–505.; Sen MK, Mahns DA, Coorssen JR, Shortland PJ. Behavioural phenotypes in the cuprizone model of central nervous system demyelination. Neurosci Biobehav Rev 2019;107:23–46.; Zhan J, Mann T, Joost S, Behrangi N, Frank M, Kipp M. The Cuprizone Model: Dos and Do Nots. Cells 2020;9.). Furthermore, during the immunohistochemical analyses, due to the large amount of coronal section obtained from different regions (0.14, -0.22, -1.06, and -1.94 mm) of the brain regions (according to the mouse brain atlas of Paxinos and Franklin (2001)), we able to perform the luxol fast blue/ cresil violet staining in several stages, but each time we also stained control and cuprizone treated groups. The darker and lighter images may be due to the slightly diverse differentiation of lithium carbonate, but hopefully this does not greatly effect the visual presentation of the extent of demyelination and remyelination in the corpus callosum area.

Thank you for your very important observation, we have improperly indicated the method, which we used for the evaluation of the immunohistochemical measurements in the manuscript.

For the evaluation of LFB and MBP immunohistochemical staining, we have used the AxioVision 4.8 software (Carl Zeiss Microscopy, Germany), which measured the mean intensity of different color channels on a scale from 0 to 65536. In our case, the low intensity value characterizes the higher myelin content in the control (CO) group as there was a higher tissue staining. On the other hand, the higher intensity measured in the cuprizone treated (DEM) group shows a lower rate in tissue staining resulting in a decreased myelin content.

The authors are grateful for the comment, and corrected the manuscript accordingly. Density was corrected to intensity in Figure 5 and in the relevant parts of the main text, as follows:

” LFB/CV staining for determination of myelin content by intensity measurement.”

We completed the lines 240-243 with the following information: “ To determine the myelin content, we have designated the corpus callosum based on the mouse brain atlas of Paxinos and Franklin (2001), and intensity measurement by luxol fast blue - crystal violet staining was used to determine myelin content cells in the corpus callosum.”

  • In Fig. 7 of the submitted manuscript, the authors show a significant increase in levels of TRP in the plasma. This was not found in their previous paper. The authors should explain this discrepancy. Similarly, the authors now show a significant increase in levels of TRP in week 5 (which is equivalent to their previous demyelination time point) in the CNS, while no differences were observed in their previous publication. In addition, the authors should comment has to why the absolute levels observed in their previous article is much lower than in this paper.

Studies involving the kynurenine pathway of TRP breakdown are not yet available in cuprizone model. Our research group is the first to investigate this area. In clinical examination, significantly higher TRP levels were reported among patients with multiple sclerosis compared to the control (Negrotto, L., Correale, J., Amino Acid Catabolism in Multiple Sclerosis Affects Immune Homeostasis. The Journal of Immunology 2017;198, 1900–1909.). However, literature data regarding TRP concentration are contradictory (see the detailed review by Fathi, M., Vakili, K., Yaghoobpoor, S., Tavasol, A., Jazi, K., Mohamadkhani, A., Klegeris, A., McElhinney, A., Mafi, Z., Hajiesmaeili, M., Sayehmiri, F., Dynamic changes in kynurenine pathway metabolites in multiple sclerosis: A systematic review. Front Immunol 2022;13, 1013784.).

Nevertheless, we would mention the limitations and differences in the application of distinct methods. The limitations of our study is primarily that, in the present study, we could not reproduce the level of TRP, as well as the KYNA concentration in the brain regions, as in the previous study with HPLC analysis. The difference may be due to different sample preparation (precipitation with perchloric acid vs. acetonitrile) and different detection methods (UV-VIS vs. UHPLC-MS/MS). Interestingly, the applied UHPLCMS/MS method measures KYNA in a much narrower concentration range from brain samples, compared to measurements with fluorescent detectors (Sadok I, Gamian A, Staniszewska MM. Chromatographic analysis of tryptophan metabolites. J Sep Sci 2017;40:3020–45.).

Recently, more bioanalytical measurements were carried out by mass spectrometry analysis in advantage of its outstanding selectivity, sensitivity, detection specificity and repro-ducibility, compared to HPLC methods. The cuprizone-induced model was repeated than the mass spectrometry tests, that had previously been validated according to the EMA guideline, were also repeated and the results confirmed the first measurement.

  • In the discussion (lines 349, 375 and 438), the authors mention they have investigated the levels of 5-HT. No data on this was provided in this manuscript. In addition, (lines 368-371) the authors write that they observed “the phagocytosis of the myelin sheath by microglia, extensive and severe axon damage, astrocytosis, and microgliosis”. This was not shown. Similarly, lines 390-392, the data was not provided. Please verify the wording used in the discussion to make sure that everything that is referenced from the literature does not appear to be from this manuscript.

Thank you for your observation. During our investigation, we measured the levels of 5-HT, tryptophan and kynurenine metabolites. In our manuscript, we did not present the data on 5-HT, because we did not find a significant difference between the cuprizone treated and control groups at any of the measurement times. In manuscript, due to the large amount of data obtained, we only showed the data of the significant changes (Lines 280-281).

Furthermore, thank you for your very important observation, indeed there has been a misinterpretation. In our study, we did not examine astrogliosis and microgliosis, only the degree of demyelination. We thought of literature data when formulating.

We revised the lines 449-452 with the following information:

„By the fifth week, we observed a large and severe demyelination in the corpus callosum of the CZP group. The significant demyelination, the phagocytosis of the myelin sheath by microglia, extensive and severe axon damage, astrocytosis, and microgliosis are also supported by literature data (Praet J, Guglielmetti C, Berneman Z, Van der Linden A, Ponsaerts P. Cellular and molecular neuropathology of the cuprizone mouse  model: clinical relevance for multiple sclerosis. Neurosci Biobehav Rev 2014;47:485–505.; Sen MK, Mahns DA, Coorssen JR, Shortland PJ. Behavioural phenotypes in the cuprizone model of central nervous system demyelination. Neurosci Biobehav Rev 2019;107:23–46.; Zhan J, Mann T, Joost S, Behrangi N, Frank M, Kipp M. The Cuprizone Model: Dos and Do Nots. Cells 2020;9.).”

We changed the lines 471-473 as follows:

„Similarly, a significant microglia/macrophage activation has been described already in the first few weeks of treatment, which persists up to five weeks during the demyelination period in the CPZ animal model (Praet J, Guglielmetti C, Berneman Z, Van der Linden A, Ponsaerts P. Cellular and molecular neuropathology of the cuprizone mouse  model: clinical relevance for multiple sclerosis. Neurosci Biobehav Rev 2014;47:485–505.; Sen MK, Mahns DA, Coorssen JR, Shortland PJ. Behavioural phenotypes in the cuprizone model of central nervous system demyelination. Neurosci Biobehav Rev 2019;107:23–46.).”

  • Lines 404-408: Why are the neuroprotective properties of KYNA discussed here if there is no difference observed in the CNS? This is could belong in the introduction.

Thank you for your observation. We added the following part to the introduction and we changed the lines 157-158 as follows: „A certain concentration of KYNA has been considered neuroprotective, scavenging the insult of reactive oxygen species [48,58].”

  • Lines 410-412: If CPZ treatment affects the “functions of KAT enzymes and thus, the level of KYNA”, why is there no differences observed in the CNS?

Thank you for your observation. Studies analyzing the enzyme function of KYN metabolites were performed in liver homogenates of normal mice, in the periphery (El-Sewedy SM, Abdel-Tawab GA, El-Zoghby SM, Zeitoun R, Mostafa MH, Shalaby ShM. Studies with tryptophan metabolites in vitro. Effect of zinc, manganese, copper and cobalt ions on kynurenine hydrolase and kynurenine aminotransferase in normal mouse liver. Biochemical Pharmacology 1974;23:2557–65). This result may explain the decrease in the level of peripheral KYNA in our investigation. As for the CNS, many factors play a role in the examination and analysis of brain regions, such as the complexity of the regions, sample preparation and its limitations and many additional different metabolites present in the CNS. The limitation of our study is primarily that in our present investigation we were not able to reproduce the marked decrease in KYNA levels in the brain regions as in the previous study using HPLC analysis. Interestingly, the applied UHPLCMS/MS method measures KYNA in a much narrower concentration range from brain samples, compared to measurements with fluorescent detectors (Sadok I, Gamian A, Staniszewska MM. Chromatographic analysis of tryptophan metabolites. J Sep Sci 2017;40:3020–45.). Recently, more bioanalytical measurements were carried out by mass spectrometry analysis, taking advantages of its preference, over HPLC methods.

We changed the lines 502-505 as follows: „CPZ intoxication may affect the enzyme functions in KYN metabolism. Indeed, it was shown that elevated copper concentration affects the function of the KAT enzymes in the periphery [85] and presumably thereby the level of KYNA.”

  • Lines 431-437: why is this mentioned? What does it bring to the manuscript apart from reminding the readers that even the authors obtained different results when doing the same experiment (difference in TRP levels compared to previous publication)?

We definitely wanted to highlight our limitations, as we could not reproduce the levels of some metabolites in the present study. Unfortunately, however, we can not compare the sample preparations and methodical detections used during our measuremens.

We completed the lines 525-526 with the following information:

„In a recent study, a significantly higher TRP level was observed among patients with MS compared to the control group (Negrotto, L., Correale, J., Amino Acid Catabolism in Multiple Sclerosis Affects Immune Homeostasis. The Journal of Immunology 2017;198, 1900–1909.).”

We added a following part to the discussion at the lines 527-536:

„The authors acknowledge the limitations of this study and different outcomes depending on analytical methods. In the present study, UHPLC-MS/MS analysis did not reproduce the TRP and KYNA concentrations in the brain regions, which was measured by HPLC method. The discrepancy may be ascribed to difference in sample preparation (precipitation with perchloric acid vs. acetonitrile) and in detection methods (UV-VIS vs. UHPLC-MS/MS). Interestingly, the UHPLC-MS/MS method measures KYNA in a much narrower concentration range from brain samples, compared to measurements with fluorescent detectors [96]. Recently, more bioanalytical measurements were carried out by mass spectrometry analysis in advantage of its outstanding selectivity, sensitivity, detection specificity, and reproducibility, compared to HPLC methods.”

  • Line 443 and line 33: CPZ-induced demyelination is a model. It is useful to study biological principles that might also be happening during progressive MS, but it’s not because we observed something in the model that it is the same in patients. The authors need to rephrase these two sentences to make it clear that CPZ-induced demyelination is not the same as progressive MS.

Thank you for your comments. Indeed, the cuprizone toxin-induced demyelination model is not the same as MS, but it shows great similarity in certain aspects. Four different MS lesions subtypes are decribed in this paper (Lucchinetti, C., Bruck, W., Parisi, J., Scheithauer, B., Rodriguez, M., Lassmann, H., Heterogeneity of multiple sclerosis lesions: implications for the pathogenesis ofdemyelination. Ann. Neurol. 2000;47, 707–717.). Pattern 1 und 2 lesions are belived to be autoimmune-mediated, while pattern 3 and 4 lesions are assumed to be primary oligodendrogliapathy, in which subtypes are defined by oligodendroglia depletion and pronounced apoptosis of the vessel as well as an intact blood brain-barrier. The common characteristic of the four subtypes is the reduction of myelin oligodendrocyte glycoprotein (MOG), myelin basic protein (MBP), myelin associated glycoprotein (MAG) or proteolipid protein (PLP), and the infiltration of macrophages and T-lymphocytes. Pattern 3 lesion show several pathological similarities to cuprizone toxin–induced changes, among others actively demyelintaion lesions, with the presence of many microglia/macrophages and only few T-cell involvement, as well as hypoxia-like injuries with sings of mitochondrial deficiency and metabolic stress, which eventually result in oligodendrocyte apoptosis (please see the detailed review by Praet, J., Guglielmetti, C., Berneman, Z., Van der Linden, A., Ponsaerts, P., Cellular and molecular neuropathology of the cuprizone mouse model: clinical relevance for multiple sclerosis. Neurosci. Biobehav. Rev. 2014;47, 485–505; and Sen, M.K., Mahns, D.A., Coorssen, J.R., Shortland, P.J., Behavioural phenotypes in the cuprizone model of central nervous system demyelination. Neurosci Biobehav Rev 2019;107, 23–46).

We completed the lines 34-35 with the following information:

„Here we show the profile of KYN metabolites in CPZ-induced mouse model in demyelination.”

We completed the lines 547-549 with the following information:

„In accordance with our previous study, here we showed the involvement of KYN metabolites in CPZ-induced animal model of demyelination, which is analogues to progressive MS in term of some markers.”

  • Overall, in the discussion, the authors fail to explain or put in context their results. As mentioned in lines 428-430 and 445-447, the levels of enzymes present in the tissue and their enzymatic activity would be needed to draw any significant conclusion from the data provided. As such, I fail to see the novelty in the dataset presented. The authors have more a less repeated their previous experiment and included more time points and a few more metabolites. They show that there are quantitative differences without explaining what it means, and they fail to draw conclusions or point to similarities with what is observed in patients, as alluded at the end of the abstract.  

The main goal of our present study was to determine the concentrations of several metabolites involved in the kynurenine pathway of tryptophan degradation, in addition to proving the harmful effect of cuprizone by immunohistochemical analysis. Another goal was to investigate the concentration change of various metabolites considered neuroprotective and neurotoxic in the periphery and CNS. This study was a long-term, complex and a large number of animals with sampling at multiple time points during and after treatment. We believe that this study provides clues to some questions, i.e. is it necessary to examine the groups at different times of cuprizone treatment and recovery (is there a difference in metabolite concentrations over the weeks), which KYN metabolites should be examined in the future (which enzymes can be affected), or is it necessary to analyze all brain regions. By answering the questions, we are planning a new experimental set-up, which may provide new information on the route. However, this is the first comprehensive study that clearly shows changes in metabolites involved in the kynurenine pathway of tryptophan metabolism during cuprizone intoxication. As these data are not yet published, we believe it is not yet time to provide information on the planned experimental set-up, which includes testing some kynurenic acid analogues that have affected the kynurenine pathway in various animal studies. Besides, during our previous investigation, we examined 4 metabolites, while in present study, we examined a total of 10 metabolite concentrations at five different times. Furthermore, in our current study, the analyzes and the detection of differences were done with a much more sensitive, precise and reliable method, which is why it is important to mention again the differences between the measurement methods.

Reviewer 2 Report

The manuscript entitled “The Tryptophan-Kynurenine Metabolic System is Suppressed in Cuprizone-Induced Model of Demyelination Simulating Progressive Multiple Sclerosis” describes the profile of kynurenine (KYN) metabolites in an animal model of MS, the well known cuprizone (CPZ)-induced mouse model, in demyelination phase, which is representative to progressive phase of MS, and also during remyelination phase of the experimental disease.

The results obtained from this animal model make authors to propose the monitoring of the KYN metabolite profile as a potential biomarker of progressive MS.

In the base of the CPZ-model disease, authors stabilize the first five weeks (focusing on weeks 1, 3, 5) after mice CPZ intoxication as the demyelinating phase of the animal disease, whereas the following four weeks (focusing on week 7 and week 9) as the remyelination phase of the disease.

The study is well designed and the presentation of the results is overall clear and well organized, however some parts of the manuscript should be improved and completed. A list is included bellow:

-          To study the demyelination authors perform histological analysis using luxol fast blue and crystal violet staining followed by densitometric analysis of the stained cells in the corpus callosum, using the microscope and the Zeiss Zen 2.6 image analysis software. It is not clear how exactly the analysis is performed. Staining intensity/area? What “LFB/CV staining means? Could you please describe in detail and include it in the manuscript?

Moreover, the images included in figure 4 are not reflecting the results obtained from the densitometric analysis and included in the graph of figure 5. Images at week 7 and week 9 show a blue color that is different from control image. Could you please explain this?

-          Pag 7, line 217, “MRM” abbreviation should be specified, it is not included in the abbreviation list.

-          Pg 8, line 252, “as a results”, change with “as a result”

-          Pg 8, lines 254-255, “…With two weeks in the recovery phase, the brain samples did not show any signs of myelin damage”, change with “…Brain samples did not show any signs of myelin damage within two weeks in the recovery phase”.

-          Figure 7, the concentration units in the Y axis is missing, please include it.

-          Pg 12, line 308, “…in the 3rd weeks of…” change with “…in the 3rd week of…”

-          Pg 12, line 310, “...which was still distinct in the 5th week of CPZ treatment and was even pronounced in the cortex and the hippocampus…”, change with “...which was still maintained in the 5th week of CPZ treatment and was even more pronounced in the cortex and the hippocampus…”

-          Pg 12, lines 379-380, “…During the remyelination phase, the differences in those metabolites became less significant and have become none eventually” change with “…During the remyelination phase, the differences in those metabolites decrease to become no significant...”

-          Pg 17, lines 422-423 “…furthermore, it can scavenge free radicals [89,92], bind the superoxide anion, and inhibits hematoxylin…”, change with “furthermore, it can scavenge free radicals [89,92], bind the superoxide anion, and inhibit hematoxylin…”

-          Pg 17, lines 433-437 “Nevertheless, it is important to mention a problem that affects many and is becoming increasingly common, namely the reproducibility crisis including using different analytical methods, which scientists have drawn attention to, and focus on the dilemma that the repetition of studies is not necessarily reproducible in animal as well as in human studies” rewrite the paragraph since it is not clear what authors mean.

-          Pg 17, line 443, change “analogues” with “analogue”

-          Has the 5-HT concentration been measured in the investigation? Because authors mention it in the text (pg 16, line 349 and 375; pg 17, line 438) but no results regarding it has been included in the manuscript.

-          Did authors try to quantify these metabolites in the mice cerebrospinal fluid (CSF)?

-          Is there any clinical study regarding the quantification of the tryptophan-kynurenine metabolites in plasma?

-          Do author think the proposed biomarkers quantification could be affordable in clinic practice?

These items could be included in the discussion.

Author Response

Reviewer #2:

The manuscript entitled “The Tryptophan-Kynurenine Metabolic System is Suppressed in Cuprizone-Induced Model of Demyelination Simulating Progressive Multiple Sclerosis” describes the profile of kynurenine (KYN) metabolites in an animal model of MS, the well known cuprizone (CPZ)-induced mouse model, in demyelination phase, which is representative to progressive phase of MS, and also during remyelination phase of the experimental disease.

The results obtained from this animal model make authors to propose the monitoring of the KYN metabolite profile as a potential biomarker of progressive MS.

In the base of the CPZ-model disease, authors stabilize the first five weeks (focusing on weeks 1, 3, 5) after mice CPZ intoxication as the demyelinating phase of the animal disease, whereas the following four weeks (focusing on week 7 and week 9) as the remyelination phase of the disease.

The study is well designed and the presentation of the results is overall clear and well organized, however some parts of the manuscript should be improved and completed. A list is included bellow:

The authors greatly appreciate your positive opinion. Here, we address our comments to solve the concerns below.

  • To study the demyelination authors perform histological analysis using luxol fast blue and crystal violet staining followed by densitometric analysis of the stained cells in the corpus callosum, using the microscope and the Zeiss Zen 2.6 image analysis software. It is not clear how exactly the analysis is performed. Staining intensity/area? What “LFB/CV staining means? Could you please describe in detail and include it in the manuscript? Moreover, the images included in figure 4 are not reflecting the results obtained from the densitometric analysis and included in the graph of figure 5. Images at week 7 and week 9 show a blue color that is different from control image. Could you please explain this?

Thank you for your very important observation. During our immunohistochemical analysis, we performed luxol fast blue staining with crezil violet secondary staining, for which we applied the FLB/CV (luxol fast blue/crezil violet) abbreviation. During the measurements, we marked the area of corpus callosum on each section, and measured its intensity. We examined the sections from each animal at a total of 4 different regions (coronally sectioned in 8 μm thickness obtained from different regions (0.14, -0.22, -1.06, and -1.94 mm) according to the mouse brain atlas of Paxinos and Franklin (2001), after that compared the average of the cuprizone treated and control groups and thus obtaining significant differences in the 3rd and 5th week of treatment.

Furhermore, we have improperly indicated the method, which we used for the evaluation of the immunohistochemical measurements in the manuscript. For the evaluation of luxol fast blue-cresil violet (LFB/CV) immunohistochemical staining, we have used the AxioVision 4.8 software (Carl Zeiss Microscopy, Germany), which measured the mean intensity of different color channels on a scale from 0 to 65536. In our case, the low intensity value characterizes the higher myelin content in the control (CO) group as there was a higher tissue staining. On the other hand, the higher intensity measured in the cuprizone treated (DEM) group shows a lower rate in tissue staining resulting in a decreased myelin content. Furthermore, during the immunohistochemical analyses, due to the large amount of coronal section, we could perform the luxol fast blue/cresil violet staining in several stages, but each time we also stained control and cuprizone treated groups. The darker and lighter images are due to the slightly diverse differentiation of lithium carbonate, but hopefully this does not greatly affect the visual presentation of the extent of demyelination and remyelination in the corpus callosum area.

The authors are grateful for your comment and corrected the manuscript accordingly. Density was corrected to intensity in Figure 5 and the relevant parts of the main text, as follows:

” LFB/CV staining for determination of myelin content by intensity measurement.”

We completed the lines 240-243 with the following information: “To determine the myelin content, we have designated the corpus callosum based on the mouse brain atlas of Paxinos and Franklin (2001), and intensity measurement by luxol fast blue - crystal violet staining was used to determine myelin content cells in the corpus callosum.”

  • Pag 7, line 217, “MRM” abbreviation should be specified, it is not included in the abbreviation list.

Thank you. We corrected the abbreviation as follows:

„MRM- Multiple Reaction Monitoring” (Line 621)

We changed the Page 8, Lines 267-269 as follows: ”Multiple Reaction Monitoring (MRM) transition of Picolinic acid was 124.0/106.0 using 75 V as declustering potential and 13 V as collision energy, retention time: 1.21 min.”

  • Page 8, Line 252, “as a results”, change with “as a result”

We changed the Page 8, Lines 302-304 as follows: ”As a result of CPZ-induced intoxication, the corpus callosum showed a significant demyelination as early as week 3 of treatment (acute demyelination phase), which damage was even more extensive in the 5th week of treatment.”

  • Pg 8, lines 254-255, “…With two weeks in the recovery phase, the brain samples did not show any signs of myelin damage”, change with “…Brain samples did not show any signs of myelin damage within two weeks in the recovery phase”.

We changed the Page 8, Lines 304-306 as follows: ” With two weeks in the recovery phase, brain samples did not show any signs of myelin damage, and nor did the samples taken in the 9th week of experiment (Figure 4. and Figure 5.).”

  • Figure 7, the concentration units in the Y axis is missing, please include it.

Thank you. The y–axis concentration units in Figure 7. are missing, because the scale is the same, but tryptophan (TRP) is uM, while the other displayed kynurenine metabolites are nM, which is why we did not display the concentration units on the y-axis.

However, in Lines 373-375, in the caption, we clarified the units for metabolites: „Figure 7. Alteration in plasma metabolite levels at week 5 of CPZ treatment. In addition to the KYNA, 3-HK, and XA (nM), we also observed a significant difference in the concentration of ANA (nM) and TRP (µM) metabolites by the 5th week of CPZ treatment between CO and CPZ groups.”

  • Page 12, Line 308, “…in the 3rd weeks of…” change with “…in the 3rd week of…”

We changed the Page 12, Lines 379-381 as follows: „However, in the 3rd week of the CPZ intoxication, in addition to the striatum, a significant reduction in 3-HK concentration was also noticed in the cortex, the hippocampus…”

  • Pg 12, line 310, “...which was still distinct in the 5th week of CPZ treatment and was even pronounced in the cortex and the hippocampus…”, change with “...which was still maintained in the 5th week of CPZ treatment and was even more pronounced in the cortex and the hippocampus…”

We changed the Page 12, Line 382-383 as follows:”… which was still maintained in the 5th week of CPZ treatment and was even more pronounced in the cortex and the hippocampus (Figure 8.).”

  • Pg 12, lines 379-380, “…During the remyelination phase, the differences in those metabolites became less significant and have become none eventually” change with “…During the remyelination phase, the differences in those metabolites decrease to become no significant...”

We changed the Page 12, Lines 460-461 as follows: „During the remyelination phase, the differences in those metabolites decrease to become no significant and have become none eventually.”

  • Pg 17, lines 422-423 “…furthermore, it can scavenge free radicals [89,92], bind the superoxide anion, and inhibits hematoxylin…”, change with “furthermore, it can scavenge free radicals [89,92], bind the superoxide anion, and inhibit hematoxylin…”

We changed the Page 17, Lines 515-517 as follows: „furthermore, it can scavenge free radicals [89,92], bind the superoxide anion, and inhibit hematoxylin autoxidation (see the review [89]).”

  • Page 17, Lines 433-437 “Nevertheless, it is important to mention a problem that affects many and is becoming increasingly common, namely the reproducibility crisis including using different analytical methods, which scientists have drawn attention to, and focus on the dilemma that the repetition of studies is not necessarily reproducible in animal as well as in human studies” rewrite the paragraph since it is not clear what authors mean.

We changed the Page 17, Lines 539-543 as follows: „Nevertheless, it is important to mention a problem that affects many and is becoming increasingly common, namely the reproducibility crisis including using different analytical methods. The scientists have drawn attention to and focus on the dilemma that the repetition of studies is not necessarily reproducible in animal as well as in human studies (97).”

  • Page 17, Line 443, change “analogues” with “analogue”

We changed the Page 17, Line 547-549 as follows: „In accordance with our previous study, here we showed the involvement of KYN metabolites in CPZ-induced animal model of demyelination, which is analogue to progressive MS in term of some markers.”

  • Has the 5-HT concentration been measured in the investigation? Because authors mention it in the text (pg 16, line 349 and 375; pg 17, line 438) but no results regarding it has been included in the manuscript.

Thank you for your observation. During our investigation, we measured the levels of 5-HT, tryptophan, and kynurenine metabolites. In our manuscript, we did not present the data on 5-HT, because we did not find a significant difference between the cuprizone treated and control groups at any of the measurement times. In manuscript, due to the large amount of data obtained, we only the significant changes were shown and published, where we observed a difference between the groups at the level of some metabolites, for the sake of a simpler review. (Lines 342-343).

  • Did authors try to quantify these metabolites in the mice cerebrospinal fluid (CSF)?

Among our plans is the examination of the cerebrospinal fluid (CSF) of mice, because so far we have not had suitable equipment to extract the CSF sample of mice. However, in the near future, we will purchase a stereotaxic equipment. Hopefully we will be able to extract the right amount of CSF from 20-25 gram mouse. Therefore, we plan to conduct a CSF analysis in the near future, as the analysis of kynurenine metabolites in the CSF of mice would be extremly informative and useful to investigate, and these results would reliably show the distribution of metabolites in the CNS.

  • Is there any clinical study regarding the quantification of the tryptophan-kynurenine metabolites in plasma?

There are several literatures available in which the amount of tryptophan-kynurenine metabolites in human plasma in various neurodegenerative (including MS) and other diseases was determined and quantified within the framework of clinical studies (Amirkhani, A., Rajda, C., Arvidsson, B., Bencsik, K., Boda, K., Seres, E., Markides, K.E., Vécsei, L., Bergquist, J., Interferon-β affects the tryptophan metabolism in multiple sclerosis patients. European Journal of Neurology 2005;12, 625–631; Hartai, Z., Klivenyi, P., Janaky, T., Penke, B., Dux, L., Vecsei, L., Kynurenine metabolism in multiple sclerosis. Acta Neurologica Scandinavica 2005;112, 93–96.; Heilman, P.L., Wang, E.W., Lewis, M.M., Krzyzanowski, S., Capan, C.D., Burmeister, A.R., Du, G., Escobar Galvis, M.L., Brundin, P., Huang, X., Brundin, L., Tryptophan Metabolites Are Associated With Symptoms and Nigral Pathology in Parkinson’s Disease. Mov Disord 2020;35, 2028–2037.; Joisten, N., Rademacher, A., Warnke, C., Proschinger, S., Schenk, A., Walzik, D., Knoop, A., Thevis, M., Steffen, F., Bittner, S., Gonzenbach, R., Kool, J., Bloch, W., Bansi, J., Zimmer, P., Exercise Diminishes Plasma Neurofilament Light Chain and Reroutes the Kynurenine Pathway in Multiple Sclerosis. Neurol Neuroimmunol Neuroinflamm 2021;8, e982.; Myint, A.-M., Kim, Y.K., Verkerk, R., Scharpé, S., Steinbusch, H., Leonard, B., Kynurenine pathway in major depression: evidence of impaired neuroprotection. J Affect Disord 2007;98, 143–151.; Giil, L.M., Midttun O., Refsum H., Ulvik A., Advani R., Smith A.D., Ueland P.M., Kynurenine Pathway Metabolites in Alzheimer’s Disease. J of Alzheimer’s Disease, 2017;vol 60, no.2, pp.495-504.)

  • Do author think the proposed biomarkers quantification could be affordable in clinic practice?

Lately, tryptophan metabolism and the kynurenine pathway have increasingly become the focus of research, especially in studies of various neurodegenerative disorders. Several studies and review have reported changes in tryptophan-kynurenine metabolite concentrations and pathway shifts, including in multiple sclerosis (see the reviews Biernacki, T., Sandi, D., Bencsik, K., Vécsei, L., Kynurenines in the Pathogenesis of Multiple Sclerosis: Therapeutic Perspectives. Cells 2020;9.; Huang, Y.-S., Ogbechi, J., Clanchy, F.I., Williams, R.O., Stone, T.W., IDO and Kynurenine Metabolites in Peripheral and CNS Disorders. Front Immunol 2020;11.; Pukoli, D., Polyák, H., Rajda, C., Vécsei, L., Kynurenines and Neurofilament Light Chain in Multiple Sclerosis. Front Neurosci 2021;15, 658202.; Sandi, D., Fricska-Nagy, Z., Bencsik, K., Vécsei, L., Neurodegeneration in Multiple Sclerosis: Symptoms of Silent Progression, Biomarkers and Neuroprotective Therapy—Kynurenines Are Important Players. Molecules 2021;26, 3423.; Vécsei, L., Szalárdy, L., Fülöp, F., Toldi, J., Kynurenines in the CNS: recent advances and new questions. Nat Rev Drug Discov 2013;12, 64–82.). Furthermore, several studies reported on the differences and relevance of the kynurenine metabolites of tryptophan breakdown in various disorders (Gaetani, L., Boscaro, F., Pieraccini, G., Calabresi, P., Romani, L., Di Filippo, M., Zelante, T.,. Host and Microbial Tryptophan Metabolic Profiling in Multiple Sclerosis. Front Immunol 2020;11, 157.; Joisten, N., Rademacher, A., Bloch, W., Schenk, A., Oberste, M., Dalgas, U., Langdon, D., Caminada, D., Purde, M.-T., Gonzenbach, R., Kool, J., Zimmer, P., Bansi, J., Influence of different rehabilitative aerobic exercise programs on (anti-) inflammatory immune signalling, cognitive and functional capacity in persons with MS – study protocol of a randomized controlled trial. BMC Neurol 2019;19, 37.; Lim, C.K., Bilgin, A., Lovejoy, D.B., Tan, V., Bustamante, S., Taylor, B.V., Bessede, A., Brew, B.J., Guillemin, G.J.. Kynurenine pathway metabolomics predicts and provides mechanistic insight into multiple sclerosis progression. Sci Rep 2017;7.; Tömösi, F., Kecskeméti, G., Cseh, E.K., Szabó, E., Rajda, C., Kormány, R., Szabó, Z., Vécsei, L., Janáky, T., A validated UHPLC-MS method for tryptophan metabolites: Application in the diagnosis of multiple sclerosis. Journal of Pharmaceutical and Biomedical Analysis 2020;185, 113246.)

In addition, our research group investigated the relationship between neurofilament light chain (NFL) neurodegenerative biomarkers and different neuroinflammation markers, including kynurenine metabolites, among patients with multiple sclerosis. We found, that the neurofilament light chain, neopterin and quinolinic acid were correlated, that is an increased concentration was found in the CSF of patients with multiple sclerosis (Rajda, C., Galla, Z., Polyák, H., Maróti, Z., Babarczy, K., Pukoli, D., Vécsei, L., Cerebrospinal Fluid Neurofilament Light Chain Is Associated with Kynurenine Pathway Metabolite Changes in Multiple Sclerosis. Int J Mol Sci 2020;21.).

Reviewer 3 Report

The authors performed an analysis of brain and peripheral metabolites of the Tryptophan-Kynurenine pathway in an animal model of Multiple sclerosis. Basically, they fed mice with cuprozine and checked several metabolites at different time points. The authors must address the following points:

1. On the abstract, the authors state: "In this study, we measured the levels of serotonin, TRP, and KYN metabolites in the plasma and the brain samples of C57Bl/6J male mice at weeks 1, 3, and 5 of demyelination, and at weeks 7 and 9 of remyelination periods by ultra-high-performance liquid chromatography with tandem mass spectrometry (UHPLC-MS/MS) after body weight measurement and immunohistochemical analysis to confirm the development of demyelination".

Key point: The authors must indicate that mice were administered with CPZ.

2. The abstract requires proofreading. Check the abstract also for typos. 

3. On Fig. 3. The body weight of CPZ-treated animals was significantly reduced for a long period of time. Was the brain weight of these animals also decreased during the same period of time? 

4. On Fig. 4. The panels must have the same contrast. Indicate the regions that were used for quantitative analysis.

5. On Fig. 5. The bars are apparently swapped. 

6. On Fig. 7. Because the concentration of TRP is in μM, it should be shown in a separate graph. This small change will avoid confusions.

7. Were the metabolite levels normalised by body weight?

8. Are the changes in the metabolite levels in plasma associated to changes in the corpus callosum?

9. Are the changes in brain metabolite levels associated to changes in the volume/weight of specific brain regions?

10. Are the effects of CPZ in this animal model associated with changes in copper levels and/or the activity of copper-enzymes that may participate in the Tryptophan-Kynurenine pathway?

Author Response

Reviewer #3:

The authors performed an analysis of brain and peripheral metabolites of the Tryptophan-Kynurenine pathway in an animal model of Multiple sclerosis. Basically, they fed mice with cuprozine and checked several metabolites at different time points. The authors must address the following points:

The authors greatly appreciate the detailed comments and suggestions of the Reviewer. We would like to answer your suggestions to the best of our knowledge.

  1. On the abstract, the authors state: "In this study, we measured the levels of serotonin, TRP, and KYN metabolites in the plasma and the brain samples of C57Bl/6J male mice at weeks 1, 3, and 5 of demyelination, and at weeks 7 and 9 of remyelination periods by ultra-high-performance liquid chromatography with tandem mass spectrometry (UHPLC-MS/MS) after body weight measurement and immunohistochemical analysis to confirm the development of demyelination".

Key point: The authors must indicate that mice were administered with CPZ.

Thank you for your observation, we changed the abstract as follows: “In this study, C57Bl/6J male mice were treated with 0.2% CPZ toxin for 5 weeks and then underwent 4 weeks of recovery. We measured the levels of serotonin, TRP, and KYN metabolites in the plasma and the brain samples of mice at weeks 1, 3, and 5 of demyelination, and at weeks 7 and 9 of remyelination periods by ultra-high-performance liquid chromatography with tandem mass spectrometry (UHPLC-MS/MS) after body weight measurement and immunohistochemical analysis to confirm the development of demyelination.”

  1. The abstract requires proofreading. Check the abstract also for typos. 

Thank you for your observation, we checked for possible typos and corrected them.

  1. On Fig. 3. The body weight of CPZ-treated animals was significantly reduced for a long period of time. Was the brain weight of these animals also decreased during the same period of time? 

In the present study, we did not measure the brain weight of the animals during cuprizone treatment and remyelination phase. However, based on literature results, the dry mass of the brain irreversibly reduced after cuprizone intoxication (Venturini, G., Enzymic activities and sodium, potassium and copper concentrations in mouse brain and liver after cuprizone treatment in vivo. J. Neurochem. 1973;21, 1147–1151.; Wakabayashi, T., Asano, M., Kurono, C.. Mechanism of the formation of megamitochondria induced by copper-chelating agents, II. Isolation and some properties of megamitochondria from the cuprizone-treated mouse liver. ActaPathol. Jpn. 1975;25, 39–49.), which reduction in brain weight was also indicated by the thinning of the corpus callosum and cortex after cuprizone treatment (Song, S.K., Yoshino, J., Le, T.Q., Lin, S.J., Sun, S.W., Cross, A.H., et al., Demyelination increases radial diffusivity in corpus callosum of mouse brain. NeuroImage 26; 2005, 132–140.; Fairless, A.H., Dow, H.C., Toledo, M.M., Malkus, K.A., Edelmann, M., Li, H., et al., Low sociability is associated with reduced size of the corpus callosum in the BALB/cJ inbred mouse strain. Brain Res. 2008;1230, 211–217.; Parenti, R., Cicirata, F., Zappala, A., Catania, A., La Delia, F., Cicirata, V., et al., Dynamic expression of Cx47 in mouse brain development and in the cuprizone model of myelin plasticity. Glia 2010;58, 1594–1609.).

  1. On Fig. 4. The panels must have the same contrast. Indicate the regions that were used for quantitative analysis.

Thank you for your observation, we have modified and standardized it the contrast of the panels in Figure 4.

During our immunohistochemical analyses, we selected the area of corpus callosum based on the mouse brain atlas of Paxinos and Franklin (2001) and measured its intensity.

We completed the Lines 240-243 with the following information: “To determine the myelin content, we selected the corpus callosum based on the mouse brain atlas of Paxinos and Franklin (2001), and intensity measurement by luxol fast blue - crystal violet staining was used to determine myelin content cells in the corpus callosum.”

  1. On Fig. 5. The bars are apparently swapped. 

Thank you for your very important observation, we have improperly indicated the method, which we used for the evaluation of the immunohistochemical measurements in the manuscript.

For the evaluation of LFB and MBP immunohistochemical staining, we have used the AxioVision 4.8 software (Carl Zeiss Microscopy, Germany), which measured the mean intensity of different color channels on a scale from 0 to 65536. In our case, the low intensity value characterizes the higher myelin content in the control (CO) group as there was a higher tissue staining. On the other hand, the higher intensity measured in the cuprizone treated (DEM) group shows a lower rate in tissue staining resulting in a decreased myelin content.

The authors are grateful for the comment and corrected the manuscript accordingly. Density was corrected to intensity in Figure 5 and in the relevant parts of the main text, as follows:

” LFB/CV staining for determination of myelin content by intensity measurement.”

We completed Lines 240-243 with the following information: “To determine the myelin content, we selected the corpus callosum based on the mouse brain atlas of Paxinos and Franklin (2001), and intensity measurement by luxol fast blue - crystal violet staining was used to determine myelin content cells in the corpus callosum.”

  1. On Fig. 7. Because the concentration of TRP is in μM, it should be shown in a separate graph. This small change will avoid confusions.

Indeed, at first glance, at the lack of concentrations and their deviations are confusing and not clear, but we wanted to show on one figure the differences in the kynurenine metabolite concentration experienced in the plasma during 5th week of cuprizone treatment. Due to the difference in the concentration units, we did not want to separate the results into two figures. Furthermore, in the description of the figure, we clearly indicate the concentration units after the metabolites.

  1. Were the metabolite levels normalised by body weight?

During the investigation, the body weight measurements showed a significant difference between the cuprizone treatment and control group. The kynurenine metabolites presented in the manuscript also showed significant differences in response to the treatment. During remyelination period, the body weight of the cuprizone treated animals gradually increased and in the 2nd week of recovery phase, we no longer observed any difference in the body weight of the groups. Metabolite measurements were also performed during 2nd week of remyelination. At the time, we no longer experienced deviations in the concetration of KYN metabolites between the groups. Based on these data, it seems that in the recovery phase the metabolite levels normalized quickly in the process of remyelination.

  1. Are the changes in the metabolite levels in plasma associated to changes in the corpus callosum?

In the first few weeks of cuprizone intoxication, the depletion of mature oligodendrocytes and oligodendrocytosis begin, which results in continuous demyelination until the end of cuprizone treatment. In the 4-5 weeks of intoxication, strong demyelination is already observed, with activation of microglia and macrophages. In this so-called acute phase, the activation of glial reactivity is accelerated by the increased degeneration of mature oligodendrocytes. Oligodendrocyte progenitor cells appear simultaneously with the degeneration of mature oligodendrocytes. After cessation of cuprizone treatment, gliosis resolves and rapid remyelination begins, when the new mature oligodendrocytes are regenerated from the oligodendrocyte progenitor cells even in the acute phase (for more details, see Sen, M.K., Mahns, D.A., Coorssen, J.R., Shortland, P.J., Behavioural phenotypes in the cuprizone model of central nervous system demyelination. Neurosci Biobehav Rev 2019; 107, 23–46). As for the plasma metabolite concentrations, already in the first week we observed a difference in the level of KYNA, 3-HK and XA between the groups, which difference persisted as the cuprizone treatment progressed. Then by the 5 week of intoxication, in addition to these metabolites, even the ANA and TRP concentrations were different. In the 2nd week of the recovery phase, we did not observe any difference in the concentration of metabolites between cuprizone treated and control groups. In other words, it seems that with remyelination, the KYN metabolite levels were also normalized.

In our study, we also performed measurements at several times during the treatment, because we wanted to investigate whether the concentrations of KYN metabolites show the differences already at the beginning of the intoxication and how the levels change during the cuprizone treatment, whether they are consistent with the demyelination and remyelination processes in the periphery and the CNS.

However, cuprizone treatment alters normal liver function (due to the megamitochondrium formation), and as a result, plasma amino acid levels also change during treatment (Goldberg, J., Daniel, M., van Heuvel, Y., Victor, M., Beyer, C., Clarner, T., et al., Short-term cuprizone feeding induces selective amino acid deprivation with concomitant activation of an integrated stress response in oligodendrocytes. Cell. Mol. Neurobiol. 2013;33, 1087–1098.).

  1. Are the changes in brain metabolite levels associated to changes in the volume/weight of specific brain regions?

In the present study, we did not measure the brain weight of the animals during cuprizone treatment and remyelination phase.

Nevertheless, cuprizone-induced oligodendrocytosis is unequally distributed in the CNS. Cuprizone intoxication causes extensive oligodendrocytosis and severe demyelination, among others in the corpus callosum, cerebral cortex, hippocampus, and to a lesser extent in the cerebellum and brainstem. The reasons for the regional variability are not known, but it may be influenced by the uneven distribution of different oligodendrocyte subtypes in the CNS, which may affect the regional variability of oligodendrocyte loss. Thus, it may happen that cuprizone is much more toxic to some subtypes, while less to others. In addition, altered gene expression can affect the sensitivity of certain areas to injuries (for more details, see Sen, M.K., Mahns, D.A., Coorssen, J.R., Shortland, P.J., Behavioural phenotypes in the cuprizone model of central nervous system demyelination. Neurosci Biobehav Rev 2019;107, 23–46).

Based on studies, the dry mass of the brain irreversibly reduced after cuprizone intoxication (Venturini, G., Enzymic activities and sodium, potassium and copper concentrations in mouse brain and liver after cuprizone treatment in vivo. J. Neurochem. 1973;21, 1147–1151.; Wakabayashi, T., Asano, M., Kurono, C.,. Mechanism of the formation of megamitochondria induced by copper-chelating agents, II. Isolation and some properties of megamitochondria from the cuprizone-treated mouse liver. ActaPathol. Jpn. 1975;25, 39–49.), which reduction in brain weight was also indicated by the thinning of the corpus callosum and cortex after cuprizone treatment (Song, S.K., Yoshino, J., Le, T.Q., Lin, S.J., Sun, S.W., Cross, A.H., et al., Demyelination increases radial diffusivity in corpus callosum of mouse brain. NeuroImage 26, 2005;132–140.; Fairless, A.H., Dow, H.C., Toledo, M.M., Malkus, K.A., Edelmann, M., Li, H., et al., Low sociability is associated with reduced size of the corpus callosum in the BALB/cJ inbred mouse strain. Brain Res. 2008;1230, 211–217.; Parenti, R., Cicirata, F., Zappala, A., Catania, A., La Delia, F., Cicirata, V., et al., Dynamic expression of Cx47 in mouse brain development and in the cuprizone model of myelin plasticity. Glia 2010;58, 1594–1609.).

In our recent study, we found a significant difference in metabolite concentration in the brain regions, that were mentioned in the literature as severely demyelinated areas including the cortex and hippocampus as well as brainstem.

Based on literature data, cuprizone also causes damage to neurotransmitter homeostasis. Cuprizone exert an inhibitory effect on glutamic acid decarboxylase, it results in a lack of energy, an increase in glutamate (GLU) level and a decrease in gamma-aminobyturic acid (GABA) (Kesterson, J.W., Carlton, W.W. Cuprizone toxicosis in mice—attempts to anti-dote the toxicity. Toxicol. Appl. Pharmacol. 1972;22, 6–13.). Another study, on the other hand, described an increased GABA level during 3 weeks of cuprizone treatment, in contrast to the reduced GABA level seen during 8 weeks of treatment, which may point to changes in neurotransmitter concentration over time during cuprizone intoxication (Biancotti, J.C., Kumar, S., de Vellis, J., Activation of inflammatory response by a combination of growth factors in cuprizone-induced demyelinated brain leads to myelin repair. Neurochem. Res. 2008;33, 2615–2628.; Praet J, Guglielmetti C, Berneman Z, Van der Linden A, Ponsaerts P. Cellular and molecular neuropathology of the cuprizone mouse model: clinical relevance for multiple sclerosis. Neurosci Biobehav Rev 2014;47:485–505).

Furthermore, based on research, dopaminergic and noradrenergic synapses are also affected during cuprizone treatment. Specifically, cuprizone poisoning has an inhibitory effect on the functioning of dopamine hydroxylase and monoamine oxidase enzymes, which affect dopamine and norepinephrine concentrations (for more details, see Praet J, Guglielmetti C, Berneman Z, Van der Linden A, Ponsaerts P. Cellular and molecular neuropathology of the cuprizone mouse model: clinical relevance for multiple sclerosis. Neurosci Biobehav Rev 2014;47:485–505), as a result, after 2 weeks of cuprizone treatment, increased dopamine and decreased norepinephrine levels were observed in the prefrontal cortex (Herring, N.R., Konradi, C., Myelin, copper, and the cuprizone model of schizophrenia. Front. Biosci. (Schol Ed) 2011;3, 23–40.; Xu, H., Yang, H.J., McConomy, B., Browning, R., Li, X.M., Behavioral and neurobiological changes in C57BL/6 mouse exposed to cuprizone: effects of antipsychotics. Front. Behav. Neurosci. 2010;4, 8.).

  1. Are the effects of CPZ in this animal model associated with changes in copper levels and/or the activity of copper-enzymes that may participate in the Tryptophan-Kynurenine pathway?

Copper, as a cofactor of various copper enzymes, plays an important role in cellular processes. Neurodegeneration may develop in case of copper homeostasis disturbance. Based on literature data, there are two hypotheses for cuprizone-induced pathology. Overall, the pathological effects of cuprizone treatment can be traced back to the disturbance of in situ copper homeostasis and the neurotoxic effect due to enzyme inhibition. However, further studies are needed to clarify these contradictory hypotheses. In order to fully explore the effect of cuprizone (for more details, see Praet J, Guglielmetti C, Berneman Z, Van der Linden A, Ponsaerts P. Cellular and molecular neuropathology of the cuprizone mouse model: clinical relevance for multiple sclerosis. Neurosci Biobehav Rev 2014;47:485–505).

Furthermore, enzymes that use copper as a cofactor are, for examples superoxid dismutase (Fridovich I. Superoxide dismutases. Annu Rev Biochem. 1975;44:147–59.), dopamine-β-hydroxylase (Blackburn NJ, Mason HS, Knowles PF. Dopamine-beta-hydroxylase: evidence for binuclear copper sites. Biochem Biophys Res Commun. 1980;95:1275–81.), monoamine oxidase (Zhang X, McIntire WS. Cloning and sequencing of a copper-containing, topa quinone-containing monoamine oxidase from human placenta. Gene. 1996;179:279–86.), the cytochrome c oxidase family (Horn D, Barrientos A. Mitochondrial copper metabolism and delivery to cytochrome c oxidase. IUBMB Life. 2008;60:421–9.), cytochrome c oxidase assembly protein (Takahashi Y, Kako K, Kashiwabara S, Takehara A, Inada Y, Arai H, Nakada K, Kodama H, Hayashi J, Baba T, Munekata E. Mammalian copper chaperone Cox17p has an essential role in activation of cytochrome C oxidase and embryonic development. Mol Cell Biol. 2002;22:7614–21; Herring, N.R., Konradi, C., Myelin, Copper, and the Cuprizone model of Schizophrenia. Front Biosci (Schol Ed) 2011;3, 23–40.).

Based on studies, the Copper Zinc Superoxide Dismutase cuproenzyme shows reduced activity when treated with cuprizone (Acs, P., Selak, M.A., Komoly, S., Kalman, B., Distribution of oligodendrocyte loss and mitochondrial toxicity in the cuprizone-induced experimental demyelination model. J. Neuroimmunol. 2013;262, 128–131.; Zhang, Y., Xu, H., Jiang, W., Xiao, L., Yan, B., He, J., et al., Quetiapine alleviates the cuprizone-induced white matter pathology in the brain of C57BL/6 mouse. Schizophr. Res. 2008;106, 182–191.; Ljutakova, S.G., Russanov, E.M., Differences in the in vivo effects of cuprizone on superoxide dismutase activity in rat liver cytosol and mitochondrial intermembrane space. Acta Physiol. Pharmacol. Bulg. 1985;11, 56–61.).

In addition, it was reported that in normal mice liver homogenates the increased copper concentration inhibited the kynurenine hydrolase and kynurenine aminotraspherase enzymes involved in the kynurenine pathway (El-Sewedy SM, Abdel-Tawab GA, El-Zoghby SM, Zeitoun R, Mostafa MH, Shalaby ShM. Studies with tryptophan metabolites in vitro. Effect of zinc, manganese, copper and cobalt ions on kynurenine hydrolase and kynurenine aminotransferase in normal mouse liver. Biochemical Pharmacology 1974;23:2557–65).

We hope that the Editor and the Reviewers will find our edited manuscript worthy of publication.

Finally, we express our gratitude to all Reviewers for their valuable remarks, criticism and constructive advice.

Round 2

Reviewer 1 Report

While the quality of the text has increased, revisions are still needed. 

The images of Figure 4 have not been changed. The differences in luminosity can impact the mean intensity of the color analyzed. I still fail to understand how a low mean intensity means a dark color and vice versa. Either the authors failed to properly explain there analysis scheme or they are mistaken. 

Unfortunately, overall, my opinion has not changed. The results provided in this manuscript is at best a small addition to their previously published data. No new conclusions are drawn or in fact, can be drawn from this data set. The authors fail to explain the discrepancies observed with the previously published data. Simply mentioning that this technique can be more precise doesn't explain the differences. Many more experiments would need to be performed to validate the new results. In my opinion, the authors need to continue this promising project and publish when they will have a complete story. I believe the enzyme levels and enzymatic activity is necessary. 

Therefore, I must once again reject this manuscript. 

Author Response

Reviewer #1:

While the quality of the text has increased, revisions are still needed.

The images of Figure 4 have not been changed. The differences in luminosity can impact the mean intensity of the color analyzed. I still fail to understand how a low mean intensity means a dark color and vice versa. Either the authors failed to properly explain there analysis scheme or they are mistaken.

Unfortunately, overall, my opinion has not changed. The results provided in this manuscript is at best a small addition to their previously published data. No new conclusions are drawn or in fact, can be drawn from this data set. The authors fail to explain the discrepancies observed with the previously published data. Simply mentioning that this technique can be more precise doesn't explain the differences. Many more experiments would need to be performed to validate the new results. In my opinion, the authors need to continue this promising project and publish when they will have a complete story. I believe the enzyme levels and enzymatic activity is necessary.

Therefore, I must once again reject this manuscript.

The authors appreciate the comments of the Reviewer. Here, we address your comments to solve the concerns below.

We revised and supplemented the text of our manuscript. Furthermore, we supplemented the images in Figure 4. with control sections at each time point, based on the suggestion of the other reviewer. In addition, at your suggestion and request, the section summarizing the body weight measurements and immunhistochemical analyzes was included as supplementary data in lines 292-297, as follows:

„ 3.1.   Investigation of body weight

The results of body weight measurement during the CPZ treatment and the recovery phase can be seen in the supplementary data.

3.2.      Evaluation of cuprizone damage in the demyelination and remyelination phases

The extent of myelin damage was examined by luxol fast blue - crystal violet staining. A detailed description and figures of the immunohistochemical analyses and subsequent intensity measurements can be found in the supplementary data.”

Supplementary data in lines: 1044-1084:

Supplementery data

  1. Results

3.1.      Investigation of body weight

On the third day of CPZ toxin treatment a significant decrease was observed in the body weight of the treated animals, compared to CO, while the decreasing tendency in the body weight disappeared upon the beginning of the remyelination phase. By the end of the investigation, both CO and treated group showed no differences regarding to the body weight of the animals (Figure 1.).

Figure 1. Alteration in body weight of the animals during the experiment. The control group is depicted with white diamonds and cuprizone treated group is depicted with white triangles. The two major parts of the experiment, i.e., the demyelination- and remyelination period are indi-cated with blue and red arrows, respectively. CO: control group; CPZ: cuprizone treated group, *: p < 0.05 vs. CO, **: p < 0.01 vs. CO, ***: p < 0.001 vs. CO. The data are presented as mean ± SEM.

3.2.      Evaluation of cuprizone damage in the demyelination and remyelination phases

The extent of myelin damage was examined by luxol fast blue - crystal violet staining.

As a result of CPZ-induced intoxication, the corpus callosum showed a significant demyelination as early as week 3 of treatment (acute demyelination phase), which dam-age was even more extensive in the 5th week of treatment. With two weeks in the recovery phase, brain samples did not show any signs of myelin damage, and nor did the samples taken in the 9th week of experiment (Figure 2. and Figure 3.)

Figure 2. Luxol fast blue - crystal violet staining in the corpus callosum of the control and cu-prizone-treated groups in the first, third, and fifth week of CPZ treatment (DEM), and in the seventh and ninth weeks of the experiment, which is the second and fourth weeks of the recov-ery phase (remyelination). A significant decrease in myelin content was observed in the third week of CPZ-treated animals (CPZ w3 vs. CO w3), compared to the CO group, which became even more pronounced by week fifth of intoxication (CPZ w5 vs. CO w5). No significant differ-ences were observed after the animals stopped receiving the CPZ (remyelination (REM) phase, CPZ w7 vs. CO w7; CPZ w9 vs. CO w9). Scale bar: 200 μm. CO: control group; CPZ: cuprizone treated group; DEM: demyelination phase in the treated group; LFB: luxol fast blue; REM: remy-elination phase in the treated group; w: week.

Figure 3. Formation of corpus callosum demyelination by CPZ treatment in the CO and CPZ group. LFB/CV staining for determination of myelin content by intensity measurement. Our re-sults show that in the third and fifth weeks of the treatment, the CPZ treatment significantly re-duced myelin content in the CPZ group compared to the CO group. CC: corpus callosum; CO: control group; CPZ: cuprizone group; LFB: luxol fast blue; W: week; **: p < 0.01 vs. CO; ***: p < 0.001 vs. CO. The data are presented as mean ± SEM.”

Regarding the explanation of the lower average intensity, demyelination occurs as a result of cuprizone treatments, as a result of which the myelin content decreases (in other animal models, as a result of various treatment, cell death occurs in this case, for explain). During the immunohistochemical analyses, the area of the corpus callosum was stained with the luxol fast blue technique. As a results of the staining, the myelin is colored blue. In the control group, the myelin content is intact, the degree of damage is not visible, which is why the fibers are markedly blue. While the animals treated with cuprizone, as a result of the treatment, de damage myelin does not stain uniformly blue, or gives a lighter color, because the myelin fibers are damaged. For this reason, it gives a more intense color value in the intensity measurements than in the control group, where the fibers were painted much darker. Therefore, our intensity measurements are higher in the cuprizone-treated group than in the control. Here we would like to mention the AxioVision 4.8 software (Carl Zeiss Microscopy, Germany) used, which measured the mean intensity of different color channels on a scale from 0 to 65536. In our case, the low intensity value characterizes the higher myelin content in the control (CO) group as there was a higher tissue staining. On the other hand, the higher intensity measured in the cuprizone treated (DEM) group shows a lower rate in tissue staining resulting in a decreased myelin content.

Furthermore, in our previous publication, we examined only 4 metabolites, while in the present study we analyzed 10 metabolites in the kynurenine pathway, including metabolites considered neuroprotective and neurotoxic. Our previous study provided only little information about the pathway, however in this study, a complete mapping of the metabolites was already done. In addition, this area has not been really researched in the literature until now, and as a results can bring us closer to the exact investigation of the deviations of the kynurenine pathway in the cuprizone model.

During the examination of the plasma, we reproduced the decrease in the kynurenic acid level. Furthermore, we also experienced additional metabolit differences, among others xanthurenic acid, tryptophan, anthranilic acid and 3-hydroxy-L-kynurenine. In the central nervous system, we also consistently experinced differences in the tryptophan and anthranilic acid and 3-hydroxy-L-kynurenine, with the fact that we could not reproduce the difference in the kynurenic acid level, which we admit is a weakness of our study.

However, the large amount of work done during the investigation and new results that we received during our analysis encourages us to present our results alongside our ongoing work. Because we believe that with this study we have extended our investigation to changes in the level of metabolites included in the pathway and to a better knowledge and understanding of it.

We trust that, based on the additions to our manuscript and our arguments, you will find the publication of our manuscript appropriate and worthy.

Reviewer 3 Report

The authors provided answers to all my questions. These answers must be reflected in the revised manuscripts. Therefore, they have to modify the manuscripts according to "author's replay". Please see below.

Thus, the authors must modify the specific sections that were mentioned in their response: Insert text and references; discuss limitations.

New comment of Fig. 4. The constrast of the sections still require work.

Moreover, they have to provide control sections for each time point.

New comment of Fig. 7. Enclose with a box the only metabolite which concentration is in micro molar. Add a note in the figure legend.

"Author's reply"

On Fig. 3. The body weight of CPZ-treated animals was significantly reduced for a long period of time. Was the brain weight of these animals also decreased during the same period of time?

In the present study, we did not measure the brain weight of the animals during cuprizone treatment and remyelination phase. However, based on literature results, the dry mass of the brain irreversibly reduced after cuprizone intoxication (Venturini, G., Enzymic activities and sodium, potassium and copper concentrations in mouse brain and liver after cuprizone treatment in vivo. J. Neurochem. 1973;21, 1147–1151.; Wakabayashi, T., Asano, M., Kurono, C.. Mechanism of the formation of megamitochondria induced by copper-chelating agents, II. Isolation and some properties of megamitochondria from the cuprizone-treated mouse liver. ActaPathol. Jpn. 1975;25, 39–49.), which reduction in brain weight was also indicated by the thinning of the corpus callosum and cortex after cuprizone treatment (Song, S.K., Yoshino, J., Le, T.Q., Lin, S.J., Sun, S.W., Cross, A.H., et al., Demyelination increases radial diffusivity in corpus callosum of mouse brain. NeuroImage 26; 2005, 132–140.; Fairless, A.H., Dow, H.C., Toledo, M.M., Malkus, K.A., Edelmann, M., Li, H., et al., Low sociability is associated with reduced size of the corpus callosum in the BALB/cJ inbred mouse strain. Brain Res. 2008;1230, 211–217.; Parenti, R., Cicirata, F., Zappala, A., Catania, A., La Delia, F., Cicirata, V., et al., Dynamic expression of Cx47 in mouse brain development and in the cuprizone model of myelin plasticity. Glia 2010;58, 1594–1609.).

On Fig. 4. The panels must have the same contrast. Indicate the regions that were used for quantitative analysis.

Thank you for your observation, we have modified and standardized it the contrast of the panels in Figure 4.

During our immunohistochemical analyses, we selected the area of corpus callosum based on the mouse brain atlas of Paxinos and Franklin (2001) and measured its intensity.

We completed the Lines 240-243 with the following information: “To determine the myelin content, we selected the corpus callosum based on the mouse brain atlas of Paxinos and Franklin (2001), and intensity measurement by luxol fast blue - crystal violet staining was used to determine myelin content cells in the corpus callosum.”

On Fig. 5. The bars are apparently swapped.

Thank you for your very important observation, we have improperly indicated the method, which we used for the evaluation of the immunohistochemical measurements in the manuscript.

For the evaluation of LFB and MBP immunohistochemical staining, we have used the AxioVision 4.8 software (Carl Zeiss Microscopy, Germany), which measured the mean intensity of different color channels on a scale from 0 to 65536. In our case, the low intensity value characterizes the higher myelin content in the control (CO) group as there was a higher tissue staining. On the other hand, the higher intensity measured in the cuprizone treated (DEM) group shows a lower rate in tissue staining resulting in a decreased myelin content.

The authors are grateful for the comment and corrected the manuscript accordingly. Density was corrected to intensity in Figure 5 and in the relevant parts of the main text, as follows:

” LFB/CV staining for determination of myelin content by intensity measurement.”

We completed Lines 240-243 with the following information: “To determine the myelin content, we selected the corpus callosum based on the mouse brain atlas of Paxinos and Franklin (2001), and intensity measurement by luxol fast blue - crystal violet staining was used to determine myelin content cells in the corpus callosum.”

On Fig. 7. Because the concentration of TRP is in μM, it should be shown in a separate graph. This small change will avoid confusions.

Indeed, at first glance, at the lack of concentrations and their deviations are confusing and not clear, but we wanted to show on one figure the differences in the kynurenine metabolite concentration experienced in the plasma during 5th week of cuprizone treatment. Due to the difference in the concentration units, we did not want to separate the results into two figures. Furthermore, in the description of the figure, we clearly indicate the concentration units after the metabolites.

Were the metabolite levels normalised by body weight?

During the investigation, the body weight measurements showed a significant difference between the cuprizone treatment and control group. The kynurenine metabolites presented in the manuscript also showed significant differences in response to the treatment. During remyelination period, the body weight of the cuprizone treated animals gradually increased and in the 2nd week of recovery phase, we no longer observed any difference in the body weight of the groups. Metabolite measurements were also performed during 2nd week of remyelination. At the time, we no longer experienced deviations in the concetration of KYN metabolites between the groups. Based on these data, it seems that in the recovery phase the metabolite levels normalized quickly in the process of remyelination.

Are the changes in the metabolite levels in plasma associated to changes in the corpus callosum?

In the first few weeks of cuprizone intoxication, the depletion of mature oligodendrocytes and oligodendrocytosis begin, which results in continuous demyelination until the end of cuprizone treatment. In the 4-5 weeks of intoxication, strong demyelination is already observed, with activation of microglia and macrophages. In this so-called acute phase, the activation of glial reactivity is accelerated by the increased degeneration of mature oligodendrocytes. Oligodendrocyte progenitor cells appear simultaneously with the degeneration of mature oligodendrocytes. After cessation of cuprizone treatment, gliosis resolves and rapid remyelination begins, when the new mature oligodendrocytes are regenerated from the oligodendrocyte progenitor cells even in the acute phase (for more details, see Sen, M.K., Mahns, D.A., Coorssen, J.R., Shortland, P.J., Behavioural phenotypes in the cuprizone model of central nervous system demyelination. Neurosci Biobehav Rev 2019; 107, 23–46). As for the plasma metabolite concentrations, already in the first week we observed a difference in the level of KYNA, 3-HK and XA between the groups, which difference persisted as the cuprizone treatment progressed. Then by the 5 week of intoxication, in addition to these metabolites, even the ANA and TRP concentrations were different. In the 2nd week of the recovery phase, we did not observe any difference in the concentration of metabolites between cuprizone treated and control groups. In other words, it seems that with remyelination, the KYN metabolite levels were also normalized.

In our study, we also performed measurements at several times during the treatment, because we wanted to investigate whether the concentrations of KYN metabolites show the differences already at the beginning of the intoxication and how the levels change during the cuprizone treatment, whether they are consistent with the demyelination and remyelination processes in the periphery and the CNS.

However, cuprizone treatment alters normal liver function (due to the megamitochondrium formation), and as a result, plasma amino acid levels also change during treatment (Goldberg, J., Daniel, M., van Heuvel, Y., Victor, M., Beyer, C., Clarner, T., et al., Short-term cuprizone feeding induces selective amino acid deprivation with concomitant activation of an integrated stress response in oligodendrocytes. Cell. Mol. Neurobiol. 2013;33, 1087–1098.).

Are the changes in brain metabolite levels associated to changes in the volume/weight of specific brain regions?

In the present study, we did not measure the brain weight of the animals during cuprizone treatment and remyelination phase.

Nevertheless, cuprizone-induced oligodendrocytosis is unequally distributed in the CNS. Cuprizone intoxication causes extensive oligodendrocytosis and severe demyelination, among others in the corpus callosum, cerebral cortex, hippocampus, and to a lesser extent in the cerebellum and brainstem. The reasons for the regional variability are not known, but it may be influenced by the uneven distribution of different oligodendrocyte subtypes in the CNS, which may affect the regional variability of oligodendrocyte loss. Thus, it may happen that cuprizone is much more toxic to some subtypes, while less to others. In addition, altered gene expression can affect the sensitivity of certain areas to injuries (for more details, see Sen, M.K., Mahns, D.A., Coorssen, J.R., Shortland, P.J., Behavioural phenotypes in the cuprizone model of central nervous system demyelination. Neurosci Biobehav Rev 2019;107, 23–46).

Based on studies, the dry mass of the brain irreversibly reduced after cuprizone intoxication (Venturini, G., Enzymic activities and sodium, potassium and copper concentrations in mouse brain and liver after cuprizone treatment in vivo. J. Neurochem. 1973;21, 1147–1151.; Wakabayashi, T., Asano, M., Kurono, C.,. Mechanism of the formation of megamitochondria induced by copper-chelating agents, II. Isolation and some properties of megamitochondria from the cuprizone-treated mouse liver. ActaPathol. Jpn. 1975;25, 39–49.), which reduction in brain weight was also indicated by the thinning of the corpus callosum and cortex after cuprizone treatment (Song, S.K., Yoshino, J., Le, T.Q., Lin, S.J., Sun, S.W., Cross, A.H., et al., Demyelination increases radial diffusivity in corpus callosum of mouse brain. NeuroImage 26, 2005;132–140.; Fairless, A.H., Dow, H.C., Toledo, M.M., Malkus, K.A., Edelmann, M., Li, H., et al., Low sociability is associated with reduced size of the corpus callosum in the BALB/cJ inbred mouse strain. Brain Res. 2008;1230, 211–217.; Parenti, R., Cicirata, F., Zappala, A., Catania, A., La Delia, F., Cicirata, V., et al., Dynamic expression of Cx47 in mouse brain development and in the cuprizone model of myelin plasticity. Glia 2010;58, 1594–1609.).

In our recent study, we found a significant difference in metabolite concentration in the brain regions, that were mentioned in the literature as severely demyelinated areas including the cortex and hippocampus as well as brainstem.

Based on literature data, cuprizone also causes damage to neurotransmitter homeostasis. Cuprizone exert an inhibitory effect on glutamic acid decarboxylase, it results in a lack of energy, an increase in glutamate (GLU) level and a decrease in gamma-aminobyturic acid (GABA) (Kesterson, J.W., Carlton, W.W. Cuprizone toxicosis in mice—attempts to anti-dote the toxicity. Toxicol. Appl. Pharmacol. 1972;22, 6–13.). Another study, on the other hand, described an increased GABA level during 3 weeks of cuprizone treatment, in contrast to the reduced GABA level seen during 8 weeks of treatment, which may point to changes in neurotransmitter concentration over time during cuprizone intoxication (Biancotti, J.C., Kumar, S., de Vellis, J., Activation of inflammatory response by a combination of growth factors in cuprizone-induced demyelinated brain leads to myelin repair. Neurochem. Res. 2008;33, 2615–2628.; Praet J, Guglielmetti C, Berneman Z, Van der Linden A, Ponsaerts P. Cellular and molecular neuropathology of the cuprizone mouse model: clinical relevance for multiple sclerosis. Neurosci Biobehav Rev 2014;47:485–505).

Furthermore, based on research, dopaminergic and noradrenergic synapses are also affected during cuprizone treatment. Specifically, cuprizone poisoning has an inhibitory effect on the functioning of dopamine hydroxylase and monoamine oxidase enzymes, which affect dopamine and norepinephrine concentrations (for more details, see Praet J, Guglielmetti C, Berneman Z, Van der Linden A, Ponsaerts P. Cellular and molecular neuropathology of the cuprizone mouse model: clinical relevance for multiple sclerosis. Neurosci Biobehav Rev 2014;47:485–505), as a result, after 2 weeks of cuprizone treatment, increased dopamine and decreased norepinephrine levels were observed in the prefrontal cortex (Herring, N.R., Konradi, C., Myelin, copper, and the cuprizone model of schizophrenia. Front. Biosci. (Schol Ed) 2011;3, 23–40.; Xu, H., Yang, H.J., McConomy, B., Browning, R., Li, X.M., Behavioral and neurobiological changes in C57BL/6 mouse exposed to cuprizone: effects of antipsychotics. Front. Behav. Neurosci. 2010;4, 8.).

Are the effects of CPZ in this animal model associated with changes in copper levels and/or the activity of copper-enzymes that may participate in the Tryptophan-Kynurenine pathway?

Copper, as a cofactor of various copper enzymes, plays an important role in cellular processes. Neurodegeneration may develop in case of copper homeostasis disturbance. Based on literature data, there are two hypotheses for cuprizone-induced pathology. Overall, the pathological effects of cuprizone treatment can be traced back to the disturbance of in situ copper homeostasis and the neurotoxic effect due to enzyme inhibition. However, further studies are needed to clarify these contradictory hypotheses. In order to fully explore the effect of cuprizone (for more details, see Praet J, Guglielmetti C, Berneman Z, Van der Linden A, Ponsaerts P. Cellular and molecular neuropathology of the cuprizone mouse model: clinical relevance for multiple sclerosis. Neurosci Biobehav Rev 2014;47:485–505).

Furthermore, enzymes that use copper as a cofactor are, for examples superoxid dismutase (Fridovich I. Superoxide dismutases. Annu Rev Biochem. 1975;44:147–59.), dopamine-β-hydroxylase (Blackburn NJ, Mason HS, Knowles PF. Dopamine-beta-hydroxylase: evidence for binuclear copper sites. Biochem Biophys Res Commun. 1980;95:1275–81.), monoamine oxidase (Zhang X, McIntire WS. Cloning and sequencing of a copper-containing, topa quinone-containing monoamine oxidase from human placenta. Gene. 1996;179:279–86.), the cytochrome c oxidase family (Horn D, Barrientos A. Mitochondrial copper metabolism and delivery to cytochrome c oxidase. IUBMB Life. 2008;60:421–9.), cytochrome c oxidase assembly protein (Takahashi Y, Kako K, Kashiwabara S, Takehara A, Inada Y, Arai H, Nakada K, Kodama H, Hayashi J, Baba T, Munekata E. Mammalian copper chaperone Cox17p has an essential role in activation of cytochrome C oxidase and embryonic development. Mol Cell Biol. 2002;22:7614–21; Herring, N.R., Konradi, C., Myelin, Copper, and the Cuprizone model of Schizophrenia. Front Biosci (Schol Ed) 2011;3, 23–40.).

Based on studies, the Copper Zinc Superoxide Dismutase cuproenzyme shows reduced activity when treated with cuprizone (Acs, P., Selak, M.A., Komoly, S., Kalman, B., Distribution of oligodendrocyte loss and mitochondrial toxicity in the cuprizone-induced experimental demyelination model. J. Neuroimmunol. 2013;262, 128–131.; Zhang, Y., Xu, H., Jiang, W., Xiao, L., Yan, B., He, J., et al., Quetiapine alleviates the cuprizone-induced white matter pathology in the brain of C57BL/6 mouse. Schizophr. Res. 2008;106, 182–191.; Ljutakova, S.G., Russanov, E.M., Differences in the in vivo effects of cuprizone on superoxide dismutase activity in rat liver cytosol and mitochondrial intermembrane space. Acta Physiol. Pharmacol. Bulg. 1985;11, 56–61.).

In addition, it was reported that in normal mice liver homogenates the increased copper concentration inhibited the kynurenine hydrolase and kynurenine aminotraspherase enzymes involved in the kynurenine pathway (El-Sewedy SM, Abdel-Tawab GA, El-Zoghby SM, Zeitoun R, Mostafa MH, Shalaby ShM. Studies with tryptophan metabolites in vitro. Effect of zinc, manganese, copper and cobalt ions on kynurenine hydrolase and kynurenine aminotransferase in normal mouse liver. Biochemical Pharmacology 1974;23:2557–65).

Author Response

All text modifications made in the main text are indicated by quotation marks in this document for transparency.

Reviewer #2:

Comments and Suggestions for Authors

The authors provided answers to all my questions. These answers must be reflected in the revised manuscripts. Therefore, they have to modify the manuscripts according to "author's replay". Please see below.

Thus, the authors must modify the specific sections that were mentioned in their response: Insert text and references; discuss limitations.

New comment of Fig. 4. The constrast of the sections still require work.

Moreover, they have to provide control sections for each time point.

New comment of Fig. 7. Enclose with a box the only metabolite which concentration is in micro molar. Add a note in the figure legend.

The authors greatly appreciate the detailed comments and suggestions of the Reviewer. Based on the suggestions, we revised, supplemented and modified the text of the manuscript.

In Figure 4. - the contrast of the images was perfected and supplemented with control sections at each time point. 

At the suggestion and request of the other reviewer, the description and figures summarizing the body weight measurements and the immunohistochemical analyses was included as supplementary data in lines 292-297, as follows:

„ 3.1.   Investigation of body weight

The results of body weight measurement during the CPZ treatment and the recovery phase can be seen in the supplementary data.

3.2.      Evaluation of cuprizone damage in the demyelination and remyelination phases

The extent of myelin damage was examined by luxol fast blue - crystal violet staining. A detailed description and figures of the immunohistochemical analyses and subsequent intensity measurements can be found in the supplementary data.”

Supplementary data in lines: 1044-1084, as follows:

Supplementary data

  1. Results

3.1.      Investigation of body weight

On the third day of CPZ toxin treatment a significant decrease was observed in the body weight of the treated animals, compared to CO, while the decreasing tendency in the body weight disappeared upon the beginning of the remyelination phase. By the end of the investigation, both CO and treated group showed no differences regarding to the body weight of the animals (Figure 1.).

Figure 1. Alteration in body weight of the animals during the experiment. The control group is depicted with white diamonds and cuprizone treated group is depicted with white triangles. The two major parts of the experiment, i.e., the demyelination- and remyelination period are indi-cated with blue and red arrows, respectively. CO: control group; CPZ: cuprizone treated group, *: p < 0.05 vs. CO, **: p < 0.01 vs. CO, ***: p < 0.001 vs. CO. The data are presented as mean ± SEM.

3.2.      Evaluation of cuprizone damage in the demyelination and remyelination phases

The extent of myelin damage was examined by luxol fast blue - crystal violet staining.

As a result of CPZ-induced intoxication, the corpus callosum showed a significant demyelination as early as week 3 of treatment (acute demyelination phase), which dam-age was even more extensive in the 5th week of treatment. With two weeks in the recovery phase, brain samples did not show any signs of myelin damage, and nor did the samples taken in the 9th week of experiment (Figure 2. and Figure 3.)

Figure 2. Luxol fast blue - crystal violet staining in the corpus callosum of the control and cu-prizone-treated groups in the first, third, and fifth week of CPZ treatment (DEM), and in the seventh and ninth weeks of the experiment, which is the second and fourth weeks of the recov-ery phase (remyelination). A significant decrease in myelin content was observed in the third week of CPZ-treated animals (CPZ w3 vs. CO w3), compared to the CO group, which became even more pronounced by week fifth of intoxication (CPZ w5 vs. CO w5). No significant differ-ences were observed after the animals stopped receiving the CPZ (remyelination (REM) phase, CPZ w7 vs. CO w7; CPZ w9 vs. CO w9). Scale bar: 200 μm. CO: control group; CPZ: cuprizone treated group; DEM: demyelination phase in the treated group; LFB: luxol fast blue; REM: remy-elination phase in the treated group; w: week.

Figure 3. Formation of corpus callosum demyelination by CPZ treatment in the CO and CPZ group. LFB/CV staining for determination of myelin content by intensity measurement. Our re-sults show that in the third and fifth weeks of the treatment, the CPZ treatment significantly re-duced myelin content in the CPZ group compared to the CO group. CC: corpus callosum; CO: control group; CPZ: cuprizone group; LFB: luxol fast blue; W: week; **: p < 0.01 vs. CO; ***: p < 0.001 vs. CO. The data are presented as mean ± SEM.”

We modified the Lines 1070-1074 with the folllowing: „A significant decrease in myelin content was observed in the third week of CPZ-treated animals (CPZ w3 vs. CO w3), compared to the CO group, which became even more pronounced by week fifth of intoxication (CPZ w5 vs. CO w5). No significant differences were observed after the animals stopped receiving the CPZ (remyelination (REM) phase, CPZ w7 vs. CO w7; CPZ w9 vs. CO w9).”

In Figure 4. –The values for the tryptophan metabolite are shown separately in the upper right part of the Figure 4., which has been changed to „B”, and detailed in the figure legend, as follows:Figure 7. Alteration in plasma metabolite levels at week 5 of CPZ treatment. In addition to the KYNA, 3-HK, and XA (nM), we also observed a significant difference in the concentration of ANA (nM) (Figure 4A) and TRP (µM) metabolites (Figure 4B) by the 5th week of CPZ treatment between CO and CPZ groups.”

On Fig. 3. The body weight of CPZ-treated animals was significantly reduced for a long period of time. Was the brain weight of these animals also decreased during the same period of time?

In the present study, we did not measure the brain weight of the animals during cuprizone treatment and remyelination phase. However, based on literature results, the dry mass of the brain irreversibly reduced after cuprizone intoxication (Venturini, G., Enzymic activities and sodium, potassium and copper concentrations in mouse brain and liver after cuprizone treatment in vivo. J. Neurochem. 1973;21, 1147–1151.; Wakabayashi, T., Asano, M., Kurono, C.. Mechanism of the formation of megamitochondria induced by copper-chelating agents, II. Isolation and some properties of megamitochondria from the cuprizone-treated mouse liver. ActaPathol. Jpn. 1975;25, 39–49.), which reduction in brain weight was also indicated by the thinning of the corpus callosum and cortex after cuprizone treatment (Song, S.K., Yoshino, J., Le, T.Q., Lin, S.J., Sun, S.W., Cross, A.H., et al., Demyelination increases radial diffusivity in corpus callosum of mouse brain. NeuroImage 26; 2005, 132–140.; Fairless, A.H., Dow, H.C., Toledo, M.M., Malkus, K.A., Edelmann, M., Li, H., et al., Low sociability is associated with reduced size of the corpus callosum in the BALB/cJ inbred mouse strain. Brain Res. 2008;1230, 211–217.; Parenti, R., Cicirata, F., Zappala, A., Catania, A., La Delia, F., Cicirata, V., et al., Dynamic expression of Cx47 in mouse brain development and in the cuprizone model of myelin plasticity. Glia 2010;58, 1594–1609.).

We modified the Lines 122-131 with the following:

„Based on literature, the weight of the brain decreases after CPZ treatment, which reduction can be explained by the thinning of the corpus callosum and cortex (see detailed Praet J, Guglielmetti C, Berneman Z, Van der Linden A, Ponsaerts P. Cellular and molecular neuropathology of the cuprizone mouse model: clinical relevance for multiple sclerosis. Neurosci Biobehav Rev 2014;47:485–505).”

We modified the Lines 559-564 with the following:

„Furthermore, based on studies, the dry mass of the brain irreversibly reduced after CPZ intoxication (Venturini, G., Enzymic activities and sodium, potassium and copper con-centrations in mouse brain and liver after cuprizone treatment in vivo. J. Neurochem. 1973;21, 1147–1151.; Wakabayashi, T., Asano, M., Kurono, C.,. Mechanism of the formation of megamitochondria induced by copper-chelating agents, II. Isolation and some properties of megamitochondria from the cuprizone-treated mouse liver. ActaPathol. Jpn. 1975;25, 39–49.), which was also indicated by the thinning of the corpus callosum and cortex after cuprizone treatment (Song, S.K., Yoshino, J., Le, T.Q., Lin, S.J., Sun, S.W., Cross, A.H., et al., Demyelination increases radial diffusivity in corpus callosum of mouse brain. NeuroImage 26, 2005;132–140.; Fairless, A.H., Dow, H.C., Toledo, M.M., Malkus, K.A., Edelmann, M., Li, H., et al., Low sociability is associated with reduced size of the corpus callosum in the BALB/cJ inbred mouse strain. Brain Res. 2008;1230, 211–217.; Parenti, R., Cicirata, F., Zappala, A., Catania, A., La Delia, F., Cicirata, V., et al., Dynamic expression of Cx47 in mouse brain development and in the cuprizone model of myelin plasticity. Glia 2010;58, 1594–1609.; Praet J, Guglielmetti C, Berneman Z, Van der Linden A, Ponsaerts P. Cellular and molecular neuropathology of the cuprizone mouse model: clinical relevance for multiple sclerosis. Neurosci Biobehav Rev 2014;47:485–505). In our study, we found a significant difference in metabolite concentration in the brain regions, that were mentioned in the literature as severely demyelinated areas, including the cortex and hippocampus as well as brainstem.”

On Fig. 4. The panels must have the same contrast. Indicate the regions that were used for quantitative analysis.

Thank you for your observation, we have modified and standardized it the contrast of the panels in Figure 4.

During our immunohistochemical analyses, we selected the area of corpus callosum based on the mouse brain atlas of Paxinos and Franklin (2001) and measured its intensity.

We completed the Lines 240-243 with the following information: “To determine the myelin content, we selected the corpus callosum based on the mouse brain atlas of Paxinos and Franklin (2001), and intensity measurement by luxol fast blue - crystal violet staining was used to determine myelin content cells in the corpus callosum.”

Based on our answers, we modified the Lines 246-249 with the following:

„To determine the myelin content, we performed luxol fast blue - crystal violet staining, then the corpus callosum was marked on each section based on the mouse brain atlas of Paxinos and Franklin (2001), and intensity measurement was used on this designated area in order to determine myelin content in the corpus callosum.”

We modified the Lines 1070-1074 with the folllowing: „A significant decrease in myelin content was observed in the third week of CPZ-treated animals (CPZ w3 vs. CO w3), compared to the CO group, which became even more pronounced by week fifth of intoxication (CPZ w5 vs. CO w5). No significant differences were observed after the animals stopped receiving the CPZ (remyelination (REM) phase, CPZ w7 vs. CO w7; CPZ w9 vs. CO w9).”

On Fig. 5. The bars are apparently swapped.

Thank you for your very important observation, we have improperly indicated the method, which we used for the evaluation of the immunohistochemical measurements in the manuscript.

For the evaluation of LFB and MBP immunohistochemical staining, we have used the AxioVision 4.8 software (Carl Zeiss Microscopy, Germany), which measured the mean intensity of different color channels on a scale from 0 to 65536. In our case, the low intensity value characterizes the higher myelin content in the control (CO) group as there was a higher tissue staining. On the other hand, the higher intensity measured in the cuprizone treated (DEM) group shows a lower rate in tissue staining resulting in a decreased myelin content.

The authors are grateful for the comment and corrected the manuscript accordingly. Density was corrected to intensity in Figure 5 and in the relevant parts of the main text, as follows:

” LFB/CV staining for determination of myelin content by intensity measurement.”

We completed Lines 240-243 with the following information: “To determine the myelin content, we selected the corpus callosum based on the mouse brain atlas of Paxinos and Franklin (2001), and intensity measurement by luxol fast blue - crystal violet staining was used to determine myelin content cells in the corpus callosum.”

We modified the Lines 238-249 with the following:

”For measurements of the stained sections were taken using a Zeiss AxioImager M2 microscope, supplied with an AxioCam MRc Rev. 3 camera (Carl Zeiss Microscopy). Zeiss Zen 2.6 (blue edition)® image analysis software program were applied, which measured the mean intensity of different color channels on a scale from 0 to 65536. In our case, the low intensity value characterizes the higher myelin content in the control (CO) group as there was a higher tissue staining. On the other hand, the higher intensity measured in the cuprizone treated (DEM) group shows a lower rate in tissue staining resulting in a decreased myelin content. To determine the myelin content, we performed luxol fast blue - crystal violet staining, then the corpus callosum was marked on each section based on the mouse brain atlas of Paxinos and Franklin (2001), and intensity measurement was used on this designated area in order to determine myelin content in the corpus callosum.”

 We completed Line 1079, in Figure 3. with the following information: ” LFB/CV staining for determination of myelin content by intensity measurement.”

On Fig. 7. Because the concentration of TRP is in μM, it should be shown in a separate graph. This small change will avoid confusions.

Indeed, at first glance, at the lack of concentrations and their deviations are confusing and not clear, but we wanted to show on one figure the differences in the kynurenine metabolite concentration experienced in the plasma during 5th week of cuprizone treatment. Due to the difference in the concentration units, we did not want to separate the results into two figures. Furthermore, in the description of the figure, we clearly indicate the concentration units after the metabolites.

In accordance with the suggestion, we separately represented the TRP concentrations between the groups in Figure 4., with the designated Figure 4B.

We completed the lines 403-406 with the following information:

„Figure 4. Alteration in plasma metabolite levels at week 5 of CPZ treatment. In addition to the KYNA, 3-HK, and XA (nM), we also observed a significant difference in the concentration of ANA (nM) (Figure 4A) and TRP (µM) metabolites (Figure 4B) by the 5th week of CPZ treatment between CO and CPZ groups”

Were the metabolite levels normalised by body weight?

During the investigation, the body weight measurements showed a significant difference between the cuprizone treatment and control group. The kynurenine metabolites presented in the manuscript also showed significant differences in response to the treatment. During remyelination period, the body weight of the cuprizone treated animals gradually increased and in the 2nd week of recovery phase, we no longer observed any difference in the body weight of the groups. Metabolite measurements were also performed during 2nd week of remyelination. At the time, we no longer experienced deviations in the concetration of KYN metabolites between the groups. Based on these data, it seems that in the recovery phase the metabolite levels normalized quickly in the process of remyelination.

Based on our answers, we completed the Lines 514-528 with the following:

„In addition to our immunohistochemical analyses, we found differences in both weight and TRP-KYN metabolite levels as a result of CPZ-treatment in the demyelination phase. However, in the recovery phase these differences between the groups disappeared and it seems that parallel to body weights, metabolite concentrations also normalized relatively quickly in the remyelination process.”

Are the changes in the metabolite levels in plasma associated to changes in the corpus callosum?

In the first few weeks of cuprizone intoxication, the depletion of mature oligodendrocytes and oligodendrocytosis begin, which results in continuous demyelination until the end of cuprizone treatment. In the 4-5 weeks of intoxication, strong demyelination is already observed, with activation of microglia and macrophages. In this so-called acute phase, the activation of glial reactivity is accelerated by the increased degeneration of mature oligodendrocytes. Oligodendrocyte progenitor cells appear simultaneously with the degeneration of mature oligodendrocytes. After cessation of cuprizone treatment, gliosis resolves and rapid remyelination begins, when the new mature oligodendrocytes are regenerated from the oligodendrocyte progenitor cells even in the acute phase (for more details, see Sen, M.K., Mahns, D.A., Coorssen, J.R., Shortland, P.J., Behavioural phenotypes in the cuprizone model of central nervous system demyelination. Neurosci Biobehav Rev 2019; 107, 23–46). As for the plasma metabolite concentrations, already in the first week we observed a difference in the level of KYNA, 3-HK and XA between the groups, which difference persisted as the cuprizone treatment progressed. Then by the 5 week of intoxication, in addition to these metabolites, even the ANA and TRP concentrations were different. In the 2nd week of the recovery phase, we did not observe any difference in the concentration of metabolites between cuprizone treated and control groups. In other words, it seems that with remyelination, the KYN metabolite levels were also normalized.

In our study, we also performed measurements at several times during the treatment, because we wanted to investigate whether the concentrations of KYN metabolites show the differences already at the beginning of the intoxication and how the levels change during the cuprizone treatment, whether they are consistent with the demyelination and remyelination processes in the periphery and the CNS.

However, cuprizone treatment alters normal liver function (due to the megamitochondrium formation), and as a result, plasma amino acid levels also change during treatment (Goldberg, J., Daniel, M., van Heuvel, Y., Victor, M., Beyer, C., Clarner, T., et al., Short-term cuprizone feeding induces selective amino acid deprivation with concomitant activation of an integrated stress response in oligodendrocytes. Cell. Mol. Neurobiol. 2013;33, 1087–1098.).

Based on our answers, we completed the Lines 500-514 with the following:

„In this study, by performing immunohistochemical analyses and plasma metabolite concentration measurements performed at different times of CPZ treatment and recovery phase, we wanted to investigate, whether the concentration of KYN metabolites show the differences already at the beginning of the intoxication and how the levels change during the CPZ treatment, whether they are consistent with the demyelination and remyelination processes in the periphery and the central nervous system. Since at the beginning of CPZ poisoning, oligodendrocytosis starts already, resulting in demyelination, and later microglia and macrophage activation occurs (for more details, see Sen, M.K., Mahns, D.A., Coorssen, J.R., Shortland, P.J., Behavioural phenotypes in the cuprizone model of central nervous system demyelination. Neurosci Biobehav Rev 2019; 107, 23–46). During the plasma examination, already in the first week of the CPZ treatment, we observed significant concentration differences for certain metabolites between the groups, which became even more evident by the 5th week, but by the 2nd week of recovery, the differences has disappeared and it seems that the levels of KYN metabolites had normalized by remyelination process.

However, CPZ treatment alters normal liver function due to the megamitochondrium formation, and as a result, plasma amino acid levels also change during treatment (Goldberg, J., Daniel, M., van Heuvel, Y., Victor, M., Beyer, C., Clarner, T., et al., Short-term cuprizone feeding induces selective amino acid deprivation with concomitant activation of an integrated stress response in oligodendrocytes. Cell. Mol. Neurobiol. 2013;33, 1087–1098.).”

Are the changes in brain metabolite levels associated to changes in the volume/weight of specific brain regions?

In the present study, we did not measure the brain weight of the animals during cuprizone treatment and remyelination phase.

Nevertheless, cuprizone-induced oligodendrocytosis is unequally distributed in the CNS. Cuprizone intoxication causes extensive oligodendrocytosis and severe demyelination, among others in the corpus callosum, cerebral cortex, hippocampus, and to a lesser extent in the cerebellum and brainstem. The reasons for the regional variability are not known, but it may be influenced by the uneven distribution of different oligodendrocyte subtypes in the CNS, which may affect the regional variability of oligodendrocyte loss. Thus, it may happen that cuprizone is much more toxic to some subtypes, while less to others. In addition, altered gene expression can affect the sensitivity of certain areas to injuries (for more details, see Sen, M.K., Mahns, D.A., Coorssen, J.R., Shortland, P.J., Behavioural phenotypes in the cuprizone model of central nervous system demyelination. Neurosci Biobehav Rev 2019;107, 23–46).

Based on studies, the dry mass of the brain irreversibly reduced after cuprizone intoxication (Venturini, G., Enzymic activities and sodium, potassium and copper concentrations in mouse brain and liver after cuprizone treatment in vivo. J. Neurochem. 1973;21, 1147–1151.; Wakabayashi, T., Asano, M., Kurono, C.,. Mechanism of the formation of megamitochondria induced by copper-chelating agents, II. Isolation and some properties of megamitochondria from the cuprizone-treated mouse liver. ActaPathol. Jpn. 1975;25, 39–49.), which reduction in brain weight was also indicated by the thinning of the corpus callosum and cortex after cuprizone treatment (Song, S.K., Yoshino, J., Le, T.Q., Lin, S.J., Sun, S.W., Cross, A.H., et al., Demyelination increases radial diffusivity in corpus callosum of mouse brain. NeuroImage 26, 2005;132–140.; Fairless, A.H., Dow, H.C., Toledo, M.M., Malkus, K.A., Edelmann, M., Li, H., et al., Low sociability is associated with reduced size of the corpus callosum in the BALB/cJ inbred mouse strain. Brain Res. 2008;1230, 211–217.; Parenti, R., Cicirata, F., Zappala, A., Catania, A., La Delia, F., Cicirata, V., et al., Dynamic expression of Cx47 in mouse brain development and in the cuprizone model of myelin plasticity. Glia 2010;58, 1594–1609.).

In our recent study, we found a significant difference in metabolite concentration in the brain regions, that were mentioned in the literature as severely demyelinated areas including the cortex and hippocampus as well as brainstem.

Based on literature data, cuprizone also causes damage to neurotransmitter homeostasis. Cuprizone exert an inhibitory effect on glutamic acid decarboxylase, it results in a lack of energy, an increase in glutamate (GLU) level and a decrease in gamma-aminobyturic acid (GABA) (Kesterson, J.W., Carlton, W.W. Cuprizone toxicosis in mice—attempts to anti-dote the toxicity. Toxicol. Appl. Pharmacol. 1972;22, 6–13.). Another study, on the other hand, described an increased GABA level during 3 weeks of cuprizone treatment, in contrast to the reduced GABA level seen during 8 weeks of treatment, which may point to changes in neurotransmitter concentration over time during cuprizone intoxication (Biancotti, J.C., Kumar, S., de Vellis, J., Activation of inflammatory response by a combination of growth factors in cuprizone-induced demyelinated brain leads to myelin repair. Neurochem. Res. 2008;33, 2615–2628.; Praet J, Guglielmetti C, Berneman Z, Van der Linden A, Ponsaerts P. Cellular and molecular neuropathology of the cuprizone mouse model: clinical relevance for multiple sclerosis. Neurosci Biobehav Rev 2014;47:485–505).

Furthermore, based on research, dopaminergic and noradrenergic synapses are also affected during cuprizone treatment. Specifically, cuprizone poisoning has an inhibitory effect on the functioning of dopamine hydroxylase and monoamine oxidase enzymes, which affect dopamine and norepinephrine concentrations (for more details, see Praet J, Guglielmetti C, Berneman Z, Van der Linden A, Ponsaerts P. Cellular and molecular neuropathology of the cuprizone mouse model: clinical relevance for multiple sclerosis. Neurosci Biobehav Rev 2014;47:485–505), as a result, after 2 weeks of cuprizone treatment, increased dopamine and decreased norepinephrine levels were observed in the prefrontal cortex (Herring, N.R., Konradi, C., Myelin, copper, and the cuprizone model of schizophrenia. Front. Biosci. (Schol Ed) 2011;3, 23–40.; Xu, H., Yang, H.J., McConomy, B., Browning, R., Li, X.M., Behavioral and neurobiological changes in C57BL/6 mouse exposed to cuprizone: effects of antipsychotics. Front. Behav. Neurosci. 2010;4, 8.).

We completed the Lines 620-649 with the following information:

„The metabolite concentration differences observed in individual brain regions may be related to changes in the volume of certain regions.

CPZ-induced oligodendrocytosis is unequally distributed in the CNS. CPZ poisoning causes extensive oligodendrocytosis and severe demyelination, among others in the corpus callosum, cerebral cortex, hippocampus, and to a lesser extent in the cerebellum and brainstem. The reasons for the regional variability are not known, but it may be influenced by the uneven distribution of different oligodendrocyte subtypes in the CNS. Thus, it may happen that CPZ is much more toxic to some subtypes, while less to others. In addition, altered gene expression can affect the sensitivity of certain areas to injuries (for more details, see Sen, M.K., Mahns, D.A., Coorssen, J.R., Shortland, P.J., Behavioural phenotypes in the cuprizone model of central nervous system demyelination. Neurosci Biobehav Rev 2019;107, 23–46).

 Furthermore, based on studies, the dry mass of the brain irreversibly reduced after CPZ intoxication (Venturini, G., Enzymic activities and sodium, potassium and copper concentrations in mouse brain and liver after cuprizone treatment in vivo. J. Neurochem. 1973;21, 1147–1151.; Wakabayashi, T., Asano, M., Kurono, C.,. Mechanism of the formation of megamitochondria induced by copper-chelating agents, II. Isolation and some properties of megamitochondria from the cuprizone-treated mouse liver. ActaPathol. Jpn. 1975;25, 39–49.), which was also indicated by the thinning of the corpus callosum and cortex after CPZ treatment (Song, S.K., Yoshino, J., Le, T.Q., Lin, S.J., Sun, S.W., Cross, A.H., et al., Demyelination increases radial diffusivity in corpus callosum of mouse brain. NeuroImage 26, 2005;132–140.; Fairless, A.H., Dow, H.C., Toledo, M.M., Malkus, K.A., Edelmann, M., Li, H., et al., Low sociability is associated with reduced size of the corpus callosum in the BALB/cJ inbred mouse strain. Brain Res. 2008;1230, 211–217.; Parenti, R., Cicirata, F., Zappala, A., Catania, A., La Delia, F., Cicirata, V., et al., Dynamic expression of Cx47 in mouse brain development and in the cuprizone model of myelin plasticity. Glia 2010;58, 1594–1609. Praet J, Guglielmetti C, Berneman Z, Van der Linden A, Ponsaerts P. Cellular and molecular neuropathology of the cuprizone mouse model: clinical relevance for multiple sclerosis. Neurosci Biobehav Rev 2014;47:485–505.). In our study, we found a significant difference in metabolite concentration in the brain regions, that were mentioned in the literature as severely demyelinated areas, including the cortex and hippocampus as well as brainstem.

Moreover, based on literature data, CPZ also causes damage to neurotransmitter homeostasis. CPZ exert an inhibitory effect on glutamic acid decarboxylase, an increase in glutamate (GLU) level and a decrease in gamma-aminobyturic acid (GABA) (Kesterson, J.W., Carlton, W.W. Cuprizone toxicosis in mice—attempts to antidote the toxicity. Toxicol. Appl. Pharmacol. 1972;22, 6–13.). Another study, on the other hand, described an increased GABA level during 3 weeks of CPZ treatment, in contrast to the reduced GABA level seen during 8 weeks of treatment, which may point to changes in neurotransmitter concentration over time during CPZ intoxication (Biancotti, J.C., Kumar, S., de Vellis, J., Activation of inflammatory response by a combination of growth factors in cuprizone-induced demyelinated brain leads to myelin repair. Neurochem. Res. 2008;33, 2615–2628.; Praet J, Guglielmetti C, Berneman Z, Van der Linden A, Ponsaerts P. Cellular and molecular neuropathology of the cuprizone mouse model: clinical relevance for multiple sclerosis. Neurosci Biobehav Rev 2014;47:485–505).

Furthermore, based on research, dopaminergic and noradrenergic synapses are also affected during CPZ treatment. Specifically, CPZ poisoning has an inhibitory effect on the functioning of dopamine hydroxylase and monoamine oxidase enzymes, which affect dopamine and norepinephrine concentrations (for more details, see Praet J, Guglielmetti C, Berneman Z, Van der Linden A, Ponsaerts P. Cellular and molecular neuropathology of the cuprizone mouse model: clinical relevance for multiple sclerosis. Neurosci Biobehav Rev 2014;47:485–505).

Are the effects of CPZ in this animal model associated with changes in copper levels and/or the activity of copper-enzymes that may participate in the Tryptophan-Kynurenine pathway?

Copper, as a cofactor of various copper enzymes, plays an important role in cellular processes. Neurodegeneration may develop in case of copper homeostasis disturbance. Based on literature data, there are two hypotheses for cuprizone-induced pathology. Overall, the pathological effects of cuprizone treatment can be traced back to the disturbance of in situ copper homeostasis and the neurotoxic effect due to enzyme inhibition. However, further studies are needed to clarify these contradictory hypotheses. In order to fully explore the effect of cuprizone (for more details, see Praet J, Guglielmetti C, Berneman Z, Van der Linden A, Ponsaerts P. Cellular and molecular neuropathology of the cuprizone mouse model: clinical relevance for multiple sclerosis. Neurosci Biobehav Rev 2014;47:485–505).

Furthermore, enzymes that use copper as a cofactor are, for examples superoxid dismutase (Fridovich I. Superoxide dismutases. Annu Rev Biochem. 1975;44:147–59.), dopamine-β-hydroxylase (Blackburn NJ, Mason HS, Knowles PF. Dopamine-beta-hydroxylase: evidence for binuclear copper sites. Biochem Biophys Res Commun. 1980;95:1275–81.), monoamine oxidase (Zhang X, McIntire WS. Cloning and sequencing of a copper-containing, topa quinone-containing monoamine oxidase from human placenta. Gene. 1996;179:279–86.), the cytochrome c oxidase family (Horn D, Barrientos A. Mitochondrial copper metabolism and delivery to cytochrome c oxidase. IUBMB Life. 2008;60:421–9.), cytochrome c oxidase assembly protein (Takahashi Y, Kako K, Kashiwabara S, Takehara A, Inada Y, Arai H, Nakada K, Kodama H, Hayashi J, Baba T, Munekata E. Mammalian copper chaperone Cox17p has an essential role in activation of cytochrome C oxidase and embryonic development. Mol Cell Biol. 2002;22:7614–21; Herring, N.R., Konradi, C., Myelin, Copper, and the Cuprizone model of Schizophrenia. Front Biosci (Schol Ed) 2011;3, 23–40.).

Based on studies, the Copper Zinc Superoxide Dismutase cuproenzyme shows reduced activity when treated with cuprizone (Acs, P., Selak, M.A., Komoly, S., Kalman, B., Distribution of oligodendrocyte loss and mitochondrial toxicity in the cuprizone-induced experimental demyelination model. J. Neuroimmunol. 2013;262, 128–131.; Zhang, Y., Xu, H., Jiang, W., Xiao, L., Yan, B., He, J., et al., Quetiapine alleviates the cuprizone-induced white matter pathology in the brain of C57BL/6 mouse. Schizophr. Res. 2008;106, 182–191.; Ljutakova, S.G., Russanov, E.M., Differences in the in vivo effects of cuprizone on superoxide dismutase activity in rat liver cytosol and mitochondrial intermembrane space. Acta Physiol. Pharmacol. Bulg. 1985;11, 56–61.).

In addition, it was reported that in normal mice liver homogenates the increased copper concentration inhibited the kynurenine hydrolase and kynurenine aminotraspherase enzymes involved in the kynurenine pathway (El-Sewedy SM, Abdel-Tawab GA, El-Zoghby SM, Zeitoun R, Mostafa MH, Shalaby ShM. Studies with tryptophan metabolites in vitro. Effect of zinc, manganese, copper and cobalt ions on kynurenine hydrolase and kynurenine aminotransferase in normal mouse liver. Biochemical Pharmacology 1974;23:2557–65).

We completed the Lines 554-564 with the following information:

“CPZ intoxication may affect the enzyme functions in KYN metabolism. Indeed, it was shown that elevated copper concentration affects the function of the kynurenine hydrolase and kynurenine aminotraspherase enzymes enzymes in the periphery [85] and presumably thereby the level of KYNA. Probably, this may explain the decrease in KYNA levels in the periphery, as a result of CPZ treatment.

Based on studies, the Copper Zinc Superoxide Dismutase cuproenzyme also shows reduced activity when treated with CPZ (Acs, P., Selak, M.A., Komoly, S., Kalman, B., Distribution of oligodendrocyte loss and mitochondrial toxicity in the cuprizone-induced experimental demyelination model. J. Neuroimmunol. 2013;262, 128–131.; Zhang, Y., Xu, H., Jiang, W., Xiao, L., Yan, B., He, J., et al., Quetiapine alleviates the cuprizone-induced white matter pathology in the brain of C57BL/6 mouse. Schizophr. Res. 2008;106, 182–191.; Ljutakova, S.G., Russanov, E.M., Differences in the in vivo effects of cuprizone on superoxide dismutase activity in rat liver cytosol and mitochondrial intermembrane space. Acta Physiol. Pharmacol. Bulg. 1985;11, 56–61.). Copper, as a cofactor of various copper enzymes, among others superoxid dismutase (Fridovich I. Superoxide dismutases. Annu Rev Biochem. 1975;44:147–59.), dopamine-β-hydroxylase (Blackburn NJ, Mason HS, Knowles PF. Dopamine-beta-hydroxylase: evidence for binuclear copper sites. Biochem Biophys Res Commun. 1980;95:1275–81.), monoamine oxidase (Zhang X, McIntire WS. Cloning and sequencing of a copper-containing, topa quinone-containing monoamine oxidase from human placenta. Gene. 1996;179:279–86.), the cytochrome c oxidase family (Horn D, Barrientos A. Mitochondrial copper metabolism and delivery to cytochrome c oxidase. IUBMB Life. 2008;60:421–9.), cytochrome c oxidase assembly protein (Takahashi Y, Kako K, Kashiwabara S, Takehara A, Inada Y, Arai H, Nakada K, Kodama H, Hayashi J, Baba T, Munekata E. Mammalian copper chaperone Cox17p has an essential role in activation of cytochrome C oxidase and embryonic development. Mol Cell Biol. 2002;22:7614–21; Herring, N.R., Konradi, C., Myelin, Copper, and the Cuprizone model of Schizophrenia. Front Biosci (Schol Ed) 2011;3, 23–40.); plays a significant role in several cellular processes, and neurodegeneration may develop in the event of a disturbance in its homeostasis (Praet J, Guglielmetti C, Berneman Z, Van der Linden A, Ponsaerts P. Cellular and molecular neuropathology of the cuprizone mouse model: clinical relevance for multiple sclerosis. Neurosci Biobehav Rev 2014;47:485–505).”

Round 3

Reviewer 1 Report

Regarding the analysis of LFB staining, I believe the authors are showing the amount of transmitted light, not the mean intensity of a color. In this case, as the authors mention, if there's a high LFB staining (high myelin content), lower amount of transmitted light is capture. However, if the authors were analyzing the mean intensity of the blue color, it would be higher. The authors should change the legend, axis labelling and the text. 

Furthermore, the quality of the text (syntax, grammar, overall English quality) is still subpart and could be improved. 

Nevertheless, these two points do not prevent the manuscript from being published. As mentioned before, I believe that this study needs further work (more data) to warrant publication.